# CONCEPT-BASED LOCAL UNIFIED EXPLANATIONS

## ABSTRACT

There is a growing demand to combine model-agnostic explanation methods with concept-based explanations, as the former can explain models across different architectures while the latter makes the explanations more faithful and understandable to end-users. However, existing concept-based model-agnostic explanation methods are limited in scope, as they mainly focus on attribution-based explanations and lack support for richer explanation types such as sufficient conditions and counterfactuals, which limits their applicability. To bridge this gap, we propose a general framework ConLUX[1] to elevate existing local model-agnostic techniques to provide concept-based explanations. Our key insight is that we can uniformly extend existing local model-agnostic methods to provide unified concept-based explanations with large pre-trained models perturbation. We have instantiated ConLUX to provide concept-based explanations in three forms: attributions, sufficient conditions, and counterfactuals, and applied it to popular text, image, and multimodal models. Our evaluation results demonstrate that ConLUX provides explanations more faithful than state-of-the-art concept-based explanation methods, and provides richer explanation forms that satisfy various user needs.

## 1 INTRODUCTION

The widespread application of machine learning models has created a strong demand for explanation methods. In particular, as the structure of machine learning models varies and evolves rapidly, and effective closed-source models (e.g., GPT-4 (Achiam et al., 2023) and Gemini (et al., 2024)) become more prevalent, model-agnostic explanation methods show their appeal (Wang, 2024). These methods treat target models as black boxes and thus do not require access to internal model information, and provide explanations in forms including attributions, sufficient conditions, and counterfactuals, satisfying various user needs (Wang, 2024). On the other hand, providing concept-based explanations has emerged as a promising direction (Poeta et al., 2023; Sun et al., 2023), which are well-known to be more faithful and understandable to end-users (Zhang et al., 2021a; Ghorbani et al., 2019b; Kim et al., 2018; Sun et al., 2023). As a result, there is a growing demand for providing concept-based model-agnostic explanations for machine learning models.

While providing concept-based model-agnostic explanations is promising, existing methods remain limited in scope. They are largely restricted to attribution forms (Poeta et al., 2023) and lack support for richer explanation types such as sufficient conditions and counterfactuals. On the other hand, feature-level model-agnostic methods support diverse forms other than attributions, but they lack the interpretability and fidelity brought by using high-level concepts.

To bridge this gap, we propose ConLUX, a general and lightweight framework to elevate existing feature-level model-agnostic methods from feature level to concept level, providing concept-based explanations in various forms beyond attributions, as well as concept-based attribution explanations of higher fidelity. Additionally, we focus on local explanations, as they are tractable for end-users when explaining complex models used in real-world applications.

We achieve this goal by making two key observations. First, elevating existing local model-agnostic methods to the concept level does not require changing their core algorithms. Since these methods follow a common workflow (Liu & Zhang, 2025), we only need to 1) extract high-level concepts from input data and 2) perform perturbations at the concept level.

---

[1]Code available at `https://anonymous.4open.science/r/ConLUX/`

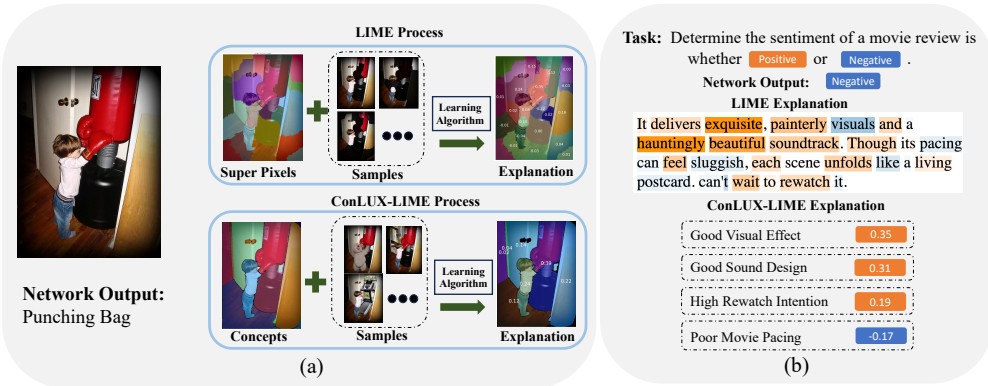

Figure 1: Examples of using LIME and ConLUX-augmented LIME to explain (a) an image classification model (YOLOv8) and (b) a text classification model (BERT). The ConLUX-augmented versions provide concept-based explanations, utilizing detected objects or topics rather than fragmented superpixels or words.

Second, given that several effective concept extraction methods already exist (Ghorbani et al., 2019a; El Shawi, 2024; Ludan et al., 2023), the only challenge is to perform perturbations at the concept level. To this end, we propose using large pre-trained models to handle concept-level perturbations.

Figure 1 shows examples of using ConLUX to elevate LIME (Ribeiro et al., 2016) to provide concept-based explanations for image and text classification models. Let us delve into the details of the image classification example. Unlike vanilla LIME using fragmented superpixels, ConLUX-augmented LIME generates explanations based on high-level concepts following Sun et al. (2023). In the perturbation phase, instead of masking fragmented superpixels, ConLUX-augmented LIME directly modifies the detected objects in the image. With this augmentation, the resulting attributions are grounded in semantically meaningful objects rather than low-level superpixels, making the explanations more interpretable for end-users.

Besides attribution-based methods, ConLUX can also extend local model-agnostic methods of various forms to provide concept-based explanations, satisfying diverse user needs and offering a more comprehensive understanding of target models. Figure 2 shows that ConLUX can extend Anchors (Ribeiro et al., 2018) and LORE (Guidotti et al., 2018) to provide concept-level rule-based sufficient conditions and counterfactual explanations, respectively. As ConLUX provides all these explanations in a unified manner, users can obtain their desired explanations by simply selecting the explanation type they need.

Moreover, although ConLUX is simple in design, it outperforms more elaborate, specially designed concept-based approaches, as our empirical evaluation results will later show.

In our evaluation, we first verified that large pre-trained models can faithfully perform concept-level perturbations. Then we evaluated ConLUX on explaining two text classification models, three image classification models, and one multimodal model. Our evaluation results show that ConLUX improves the fidelity of Anchors, LIME, LORE, and Kernel SHAP (Lundberg & Lee, 2017) explanations by 56.8%, on average, and ConLUX-augmented methods outperform state-of-the-art concept-based explanation methods specifically designed for text models (Ludan et al., 2023; Yu et al., 2024) and image models (Sun et al., 2023), respectively. We also ran a human evaluation, which shows that ConLUX helps users better use the explanations in downstream tasks.

**Contributions** Our contributions are as follows:

- We introduce ConLUX, a general and lightweight framework that elevates existing local model-agnostic explanation methods to concept level with minimal user effort, providing concept-based explanations with state-of-the-art performance, and in various forms beyond attributions.

- We propose using large pre-trained models to perform concept-level perturbations, enabling model-agnostic methods to generate concept-based explanations, and empirically verify the fidelity of these perturbations.

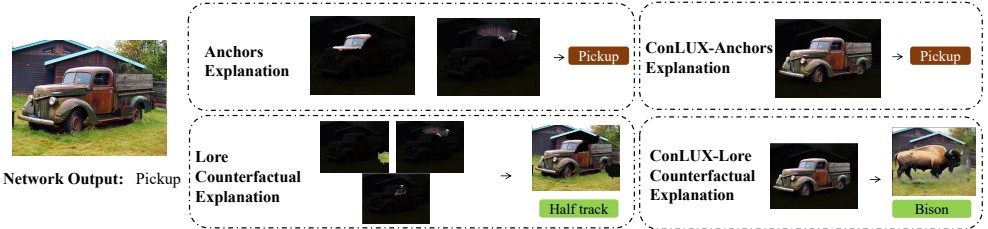

Figure 2: Example explanations from Anchors, LORE, and their ConLUX-augmented versions. The Anchors explanation states that the presence of specific image regions guarantees that the model classifies the image as a pickup. The LORE explanation shows that masking these regions would lead the model to predict a different class. The ConLUX-augmented versions provide concept-based explanations, using detected objects rather than fragmented superpixels.

- We instantiate ConLUX on four popular explanation methods: LIME, Kernel SHAP, Anchors, and LORE, and demonstrate its effectiveness through extensive experiments and human evaluations. ConLUX achieves state-of-the-art performance in generating concept-based explanations.

## 2 BACKGROUND AND RELATED WORK

Our work is related to model-agnostic explanation methods and concept-based explanation methods. We next introduce the background knowledge and related work.

### 2.1 MACHINE LEARNING MODELS

We consider a machine learning model as a function $f$ that maps an input vector $\boldsymbol{x}$ to an output scalar $f(\boldsymbol{x})$. Formally, let $f : \mathbb{X} \to \mathbb{R}$, where $\mathbb{X} = \mathbb{R}^n$. Let $\boldsymbol{x}_i$ denote the $i$-th feature value of $\boldsymbol{x}$.

### 2.2 LOCAL MODEL-AGNOSTIC EXPLANATION METHODS

A local model-agnostic explanation method $t$ takes a model $f$ and an input $\boldsymbol{x}$, and generates a local explanation $g_{f,\boldsymbol{x}}$ to describe the behavior of $f$ around $\boldsymbol{x}$, i.e., $g_{f,\boldsymbol{x}} := t(f, \boldsymbol{x})$. An explanation $g_{f,\boldsymbol{x}}$ ($g$ for short) is an expression formed with predicates. Each predicate $p$ maps an input $\boldsymbol{x}$ to a binary value, i.e., $p : \mathbb{X} \to \{0, 1\}$, indicating whether $\boldsymbol{x}$ satisfies a specific condition.

Existing local model-agnostic explanation methods follow a similar workflow:

1. **Producing Predicates**: These methods first generate a set of predicates $\mathbb{P}$ based on the input $\boldsymbol{x}$.

2. **Generating Samples**: The underlying perturbation model $t_{per}$ generates a set of samples $\mathbb{X}_s$ and its corresponding predicate representation set $\mathbb{B}_s$.

3. **Learning Explanation**: The underlying learning algorithm generates the local explanation $g_{f,\boldsymbol{x}}$ consisting of predicates in $\mathbb{P}$ using $\mathbb{B}_s$ and $f(\mathbb{X}_s)$.

Mainstream local model-agnostic explanation methods like Anchors, LIME, LORE, and Kernel SHAP, all follow this workflow. In the following, we introduce the main components of these explanation methods.

**Predicate Sets** Given an input $\boldsymbol{x}$, the corresponding predicate set $\mathbb{P}$ is defined as $\mathbb{P} = \{p_i | i \in [1, d]\}$, where $d$ is the number of predicates in $\mathbb{P}$, a hyperparameter set by users or according to the input $\boldsymbol{x}$. Each $p_i$ is a feature predicate that constrains the value of a set of feature values in $\boldsymbol{x}$, i.e. $p_i(\boldsymbol{z}) : \bigwedge_{j \in \mathbb{A}_i} \mathbb{1}_{\boldsymbol{x}_j = \boldsymbol{z}_j}$, where $\mathbb{A}_i$ is the set of indices of features that $p_i$ constrains. Then, we define the predicate representation $\boldsymbol{b} \in \{0, 1\}^d$ of a sample input $\boldsymbol{z}$ as a binary vector where $\boldsymbol{b}_i = p_i(\boldsymbol{z})$.

**Perturbation Models** The perturbation model $t_{per}$ first randomly selects $\mathbb{B}_s \subseteq \{0, 1\}^d$ as the predicate representations of the samples. Then, it transforms $\mathbb{B}_s$ back to the original input space to get $\mathbb{X}_s$. For each $\boldsymbol{b} \in \mathbb{B}_s$, if $\boldsymbol{b}_i = 1$, then for each $j \in \mathbb{A}_i$, it sets $\boldsymbol{z}_j = \boldsymbol{x}_j$; otherwise, it sets each $\boldsymbol{z}_j$ to a masked value, or a random value sampled from a user-defined distribution.

**Learning Algorithms** Existing local model-agnostic explanation methods use different learning algorithms. Anchors uses KL-LUCB (Kaufmann & Kalyanakrishnan, 2013) to learn sufficient conditions, LIME and Kernel SHAP use linear regression (McDonald, 2009) to learn attributions, and LORE uses decision trees (Ruggieri, 2004) to learn sufficient conditions and counterfactuals.

**Explanation Forms** Existing local model-agnostic explanation methods can provide various forms of explanations for different user needs:

- *Attributions*: An attribution explanation (Lundberg & Lee, 2017; Ribeiro et al., 2016; Tan et al., 2023; Shankaranarayana & Runje, 2019; Goldstein et al., 2015; Apley & Zhu, 2020) consists of a set of feature predicates $p_i$ with importance weights $w_i$, i.e., $g = \{(p_i, w_i) \mid i \in [1, d]\}$. Attributions assist users in evaluating the trustworthiness of the model (Ribeiro et al., 2016).
- *Sufficient Conditions*: This form of explanation (Ribeiro et al., 2018) identifies a minimal set of conditions that are sufficient to produce the same output as the original input. Formally, $f(z) = f(x)$ if $g(z) = 1$, where $g(z) = \bigwedge_{p \in \mathbb{Q}} p(z)$ and $\mathbb{Q} \subseteq \mathbb{P}$. Sufficient conditions help users anticipate the model's behavior on unseen inputs (Ribeiro et al., 2018).
- *Counterfactuals*: A counterfactual explanation (Wachter et al., 2018; Guidotti et al., 2018) shows how a model's prediction changes under specific input modifications. It is defined as $f(z) = y$ if $g(z) = 1$, where $y \neq f(x)$ and $g(z) = \bigwedge_{p \in \mathbb{Q}} p(z) \wedge \bigwedge_{p \in \mathbb{C}} \neg p(z)$, with $\mathbb{Q}, \mathbb{C} \subseteq \mathbb{P}$. Counterfactuals provides actionable insights for decision-making and outcome reversal (Guidotti et al., 2018).

To our knowledge, existing model-agnostic explanation methods that provide explanations other than attributions mainly provide explanations at feature levels (Zhang et al., 2021c; Wang, 2024).

### 2.3 CONCEPT-BASED EXPLANATIONS

Concept-based explanations primarily focus on attributing high-level concepts to model predictions (Poeta et al., 2023). While the definition of a concept may vary, Molnar (2020) broadly characterizes it as "an abstract idea, such as a color, an object, or even an idea," with the key requirement that concepts be meaningful to end-users (Ghorbani et al., 2019a), such as objects in images or topics in text.

Existing concept-based explanation methods have extensively explored the extraction of high-level concepts from input data, but the perturbation of high-level concepts remains largely unexplored, which is crucial for generating model-agnostic explanations. Specifically, these approaches, including various concept bottleneck models (Koh et al., 2020; Ludan et al., 2023; Oikarinen et al., 2023; Kim et al., 2023), leverage internal model representations (Zhang et al., 2021b; Yeh et al., 2019; Cunningham et al., 2023; Ghorbani et al., 2019c; Crabbé & van der Schaar, 2022; Fel et al., 2023a;b; Ghorbani et al., 2019a; Yeh et al., 2020; Varshney et al., 2025; Taparia et al.), external knowledge (El Shawi, 2024; Widmer et al., 2022; Ciravegna et al., 2023), or pre-trained models (Ludan et al., 2023; Sun et al., 2023) to extract concepts from input data. However, although Goyal et al. (2020) and Wu et al. (2022) try to map concept level changes back to feature level, they are designed for specific tasks, and need much effort to adapt to new datasets or models. Moreover, most concept-based model-agnostic methods focus primarily on attribution-based explanations, which limits their expressiveness and applicability. Although Ciravegna et al. (2023) provide rule-based sufficient conditions, they are limited to only provide global explanations, which is not tractable for complex models in practice.

## 3 THE CONLUX FRAMEWORK

In this section, we propose ConLUX, a general and lightweight framework to elevate existing local model-agnostic explanation methods to concept level without significantly changing their core components. As shown in Figure 3, ConLUX-augmented methods generate explanations in three steps: 1) concept-level predicate producing, 2) concept-level perturbation, and 3) explanation generation.

### 3.1 CONCEPT-LEVEL PREDICATE PRODUCING

We use a **concept-extracting model** to extract high-level concepts from a given input $x$ based on the target task and user needs. As existing work has proposed several effective methods for concept

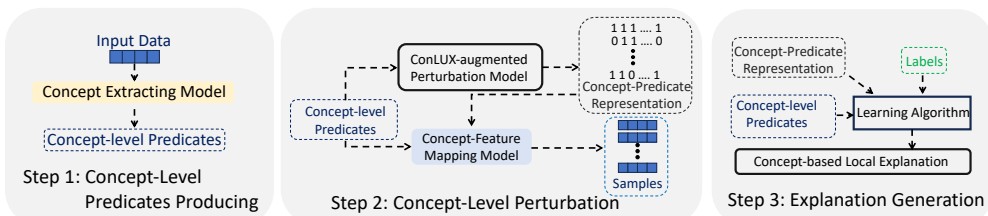

Figure 3: The workflow of ConLUX-augmented local model-agnostic explanation techniques.

extraction (Ghorbani et al., 2019a; El Shawi, 2024; Ludan et al., 2023; Sun et al., 2023), users can select an appropriate method based on their specific needs.

ConLUX defines **concept predicates** (denoted as $p^c$) based on the extracted concepts. Each concept predicate $p^c$ is a binary function that indicates whether the input $x$ satisfies a specific concept. Subsequently, ConLUX replaces the vanilla predicates set $\mathbb{P}$ with the concept predicates set $\mathbb{P}^c$.

**An Example** For the text example in Figure 1(b), our approach follows Ludan et al. (2023) to extract concepts that correlate with sentiment from a professional movie reviewer's perspective, and the model generates the four concepts. For the i-th concept, ConLUX defines its corresponding predicate $p_i^c$ as *"if the sentence has the i-th concept"*, and set the predicates set $\mathbb{P}^c = \{p_1^c, p_2^c, p_3^c, p_4^c\}$.

## 3.2 CONCEPT-LEVEL PERTURBATION

The **ConLUX augmented perturbation model** $t_{per}^c$ changes high-level concepts directly to obtain samples $\mathbb{X}_s^c$ and their concept-predicate representations $\mathbb{B}_s^c$.

Specifically, $t_{per}^c$ first generates samples in concept-predicate representations. Each **concept-predicate representation** $b^c$ is a $|\mathbb{P}^c|$-dimension binary vector corresponding to a sample $z$, indicating whether the sample satisfies the concept predicates in $\mathbb{P}^c$. Since concept predicates do not directly constrain the feature values, $t_{per}^c$ requires a **concept-feature mapping model** $c : \{0,1\}^{|\mathbb{P}^c|} \to \mathbb{X}$ to transform the samples back to the feature level. We find that large pre-trained models are ideal to serve as the concept-feature mapping model in ConLUX, as they excel at generating content based on structured prompts. Besides, their generated samples are more meaningful compared to simply masking to changing feature values (Tan et al., 2023; Ribeiro et al., 2016; 2018; Liu & Zhang, 2025). Specifically, when mapping a sample's concept-predicate representation $b^c$ back to the feature level, if $b_i^c = 1$, we prompt the model to ensure that the generated sample satisfies the i-th concept; if $b_i^c = 0$, the generated sample must not satisfy the i-th concept.

**Examples** Figure 1(a) shows some samples generated by the ConLUX-augmented perturbation model. For example, $t_{per}^c$ generated the left-upper sample without a child in the image. Appendix B shows the prompts we used to generate the samples, which does not require much effort to design.

## 3.3 EXPLANATION GENERATION

In this stage, ConLUX leverages the vanilla learning algorithms underlying existing methods to generate explanations. Using the concept-predicate representations $\mathbb{B}_s^c$ and their corresponding outputs $f(\mathbb{X}_s^c)$, the learning algorithm learns a concept-based explanation $g_{f,x}$ composed of concept predicates in $\mathbb{P}^c$. Benefiting from ConLUX's generality in augmenting various existing explanation methods, ConLUX inherits their ability to provide various forms of explanations, including attributions, sufficient conditions, and counterfactuals.

ConLUX enables the generation of multiple explanation forms through a unified framework with a single click. As a result, users can flexibly choose the explanation type that best fits their needs. We refer to the resulting collection of explanations as a **ConLUX local unified explanation**.

**Examples** Figure 1, 2 show examples of ConLUX generated attributions, sufficient conditions, and counterfactuals.

## 4 EMPIRICAL EVALUATION

We evaluate ConLUX from three aspects: (1) the fidelity of concept-level perturbation, (2) the fidelity of ConLUX-augmented explanations, and (3) how much ConLUX helps users use explanations in downstream tasks. We also discuss the time efficiency of ConLUX and the robustness of ConLUX to different generative model choices.

### 4.1 PERTURBATION FIDELITY EVALUATION

To ensure ConLUX can faithfully generate explanations, it is essential to ensure the underlying large models perform accurate concept-feature mapping.

**Experimental Setup**    We conducted experiments on two types of input data: text and image.

For text data, we use DeepSeek-V3 (Liu et al., 2024) as the concept–feature mapping model. We verify its fidelity on perturbing on both human-annotated and model-extracted concepts. We use four datasets and randomly select 250 sentences from each of the datasets. For each sentence, we instruct DeepSeek-V3 to generate 10 perturbed samples by altering specific given concepts, and using a checker to verify whether the generated sentences meet the requirements.

Specifically, for human-annotated concepts, we use two named entity recognition datasets, CoNLL-2003 (Tjong Kim Sang & De Meulder, 2003) and OntoNotes 5.0 (Pradhan et al., 2013). The concepts include entity types such as *person*, *location*, and *organization*. We use a fine-tuned BERT model (Devlin et al., 2018) as the checker. For model-extracted concepts, as Ludan et al. (2023) has verified that GPT-based models can extract high-quality textual concepts and verify concept satisfaction, we use GPT-4o to extract concepts from sentences of the Large Movie Review (Maas et al., 2011) and Fake News (Pérez-Rosas et al., 2018) datasets, and also use GPT-4o as the checker.

For image data, we used Blended Latent Diffusion (Avrahami et al., 2023) as the concept-feature mapping model, and conducted our experiment on images from the validation set of COCO dataset (Lin et al., 2014), where objects in images are annotated by humans. For each image, we generated 10 samples by altering these concepts, and checked if the generated samples satisfy the requirements by an object-detection YOLO11 model (Jocher et al., 2023).

**Metrics and Results**    We evaluated perturbation fidelity using accuracy, i.e. the proportion of generated samples that satisfy the given concept requirements. Table 1 shows that our perturbation models achieve an average accuracy of 96.8%, indicating that ConLUX can faithfully generate samples that meet the given concept requirements.

### 4.2 EXPLANATION FIDELITY EVALUATION

Our evaluation consists of two parts: (1) assessing the fidelity improvement that ConLUX brings to existing local model-agnostic explanation methods (Anchors, LIME, LORE, and Kernel SHAP (KSHAP for short)), and (2) comparing the fidelity of ConLUX with state-of-the-art concept-based methods: TBM (Ludan et al., 2023) and LACOAT (Yu et al., 2024) for text models, and EAC (Sun et al., 2023) and ConceptLIME (Tan et al., 2024) for image models.

#### 4.2.1 EXPERIMENTAL SETUP

We conducted the experiments on two text classification tasks, three image classification tasks, and one multimodal task.

**Text tasks**    Text classification models take a text sequence as input and predict its category. We used Large Movie Review (Maas et al., 2011) and Fake News (Pérez-Rosas et al., 2018) datasets, and predicted the labels of sentences in the validation set using a fine-tuned BERT (Devlin et al., 2018) and DeepSeek-V3 with in-context examples, which are the models to explain. For the baseline explanation methods, we kept the default settings. For ConLUX, we followed the settings in the perturbation fidelity evaluation.

**Image tasks**    Image classification models take an image as input and predict its category. We conducted experiments on ImageNet (Deng et al., 2009), Caltech-101 (Fei-Fei et al., 2007), and CUB-200 (Welinder et al., 2010). For each dataset, we used a fine-tuned YOLOv8, Vision Trans-

Table 1: Accuracy of using different concept-level perturbation models to generate samples that satisfy certain concept predicates. We used DeepSeekV3 and Blended Latent Diffusion in our perturbation and explanation fidelity evaluation in Section 4.1 and 4.2.

| Model | CoNLL | Onto. | Large. | Fake. | COCO |
|---|---|---|---|---|---|
| **DeepSeekV3** | 97.5 | 98.1 | 97.7 | 94.7 | – |
| Qwen2.5 72B | 98.1 | 97.3 | 97.5 | 95.0 | – |
| Qwen2.5 7B | 92.4 | 91.9 | 92.3 | 91.3 | – |
| **Blended Latent Diffusion** | – | – | – | – | 98.1 |
| Latent Consistency Model | – | – | – | – | 97.5 |

former (ViT) (Oquab et al., 2023; Darcet et al., 2023), and ResNet-50 (He et al., 2016) to classify images from the validation set and explain the models' local behaviors. For the vanilla methods, we followed LIME to use Quickshift (Jiang et al., 2018) to obtain superpixels from input images as predicates. For ConLUX, we used SAM (Kirillov et al., 2023) as the concept-extracting model, and Blended Latent Diffusion as the concept-feature mapping model. For EAC and ConceptLIME, we kept their default settings.

**Multimodal tasks**   We follow the settings in text and image tasks, explain multimodal Qwen2.5-VL on 250 randomly selected Yes/No Question of the VQAv2 (Goyal et al., 2017) dataset, and compare the fidelity of ConLUX-augmented explanations with the vanilla ones.

As TBM, LACOAT, EAC, and ConceptLIME are not applicable to multimodal tasks, we compare the fidelity of these methods and ConLUX-augmented explanations only on text and image tasks.

### 4.2.2   EVALUATION METRICS

Fidelity reflects how faithfully an explanation describes a target model. As the explanation form of these methods varies, we used different metrics to evaluate their fidelity.

For Anchors and LORE, following the settings of their original papers, we used **coverage** and **precision** as the fidelity metrics (which are named differently in the LORE paper). Given a target model $f$, an input $\boldsymbol{x}$, and a distribution $D_{\boldsymbol{x}}$ derived from the perturbation model, and the corresponding explanation $g$, we defined the coverage as $\mathrm{cov}(\boldsymbol{x}; f, g) = \mathbb{E}_{\boldsymbol{z} \sim D_{\boldsymbol{x}}}[g(\boldsymbol{z})]$, which indicates the proportion of inputs in the distribution that match the rule. We defined precision as $\mathrm{prec}(\boldsymbol{x}; f, g) = \mathbb{E}_{\boldsymbol{z} \sim D_{\boldsymbol{x}}}[\mathbf{1}_{f(\boldsymbol{z})=y}|g(\boldsymbol{z})]$, where $y$ is the consequence of the rules in $g$ with $y = f(\boldsymbol{x})$ for factual rules and $y \neq f(\boldsymbol{x})$ for counterfactual rules. Precision indicates the proportion of covered inputs for which $g$ correctly predicts the model outputs.

As LIME and KSHAP are attribution-based local surrogates, we measured their explanation fidelity by **deletion experiments**. We used *Area Over most relevant first perturbation curve* (AOPC), and $\mathrm{accuracy}_a$ as the metrics (Samek et al., 2016; Modarressi et al., 2023). Given a target model $f$, an input $\boldsymbol{x}$, the model output $y = f(\boldsymbol{x})$, their corresponding explanation $g$, and $\boldsymbol{x}^{(k)}$ that is generated by masking the $k\%$ most important predicates in $\boldsymbol{x}$, AOPC and $\mathrm{accuracy}_a$ are defined as follows:

- **AOPC:** Let $\mathrm{AOPC}_k = \frac{1}{|\mathbb{T}|} \sum_{\boldsymbol{x} \in \mathbb{T}}(p_f(y|\boldsymbol{x}) - p_f(y|\boldsymbol{x}^{(k)}))$, where $p_f(y|\boldsymbol{x})$ is the probability of $f$ to output $y$ given the input $\boldsymbol{x}$, and $\mathbb{T}$ is the set of all test inputs. $\mathrm{AOPC}_k$ indicates the average change of the model output when masking the $k\%$ most important predicates. A higher $\mathrm{AOPC}_k$ indicates a better explanation. AOPC is defined as $\mathrm{AOPC} = \sum_{k=1}^{100} \mathrm{AOPC}_k/100$.
- **Accuracy$_a$:** $\mathrm{Accuracy}_a$ indicates the proportion of inputs among all $\boldsymbol{x}^{(k)}$ that the target model gives the same output as the original input $\boldsymbol{x}$, i.e. $\mathbb{E}[f(\boldsymbol{x}^{(k)}) = f(\boldsymbol{x})]$. Specifically, $\mathrm{accuracy}_a$ is different from the standard accuracy, and a lower $\mathrm{accuracy}_a$ indicates a better explanation.

Specifically, we only considered the predicates that positively contribute to $f(\boldsymbol{x})$.

When comparing to state-of-the-art concept-based explanation methods, we considered ConLUX unified explanations as one of the main advantages of ConLUX over existing concept-based approaches is offering rich forms of explanations. Since there are multiple forms of explanations, considering that they can all serve as local surrogate models, we defined the metrics as follows (Balagopalan et al., 2022; Ismail et al., 2021): given a target model $f$, an input $\boldsymbol{x}$, a perturbation distri-

Table 2: Average coverage and precision (higher is better) of Anchors, LORE, and their ConLUX-augmented versions.

| Models | Coverage (%) ↑ | | | | Precision (%) ↑ | | | |
|---|---|---|---|---|---|---|---|---|
| | Anchors | Anchors* | LORE | LORE* | Anchors | Anchors* | LORE | LORE* |
| DeepSeek-V3/Large. | 4.7±0.7 | **23.5**±2.0 | 2.3±0.3 | **22.4**±2.1 | 79.2±1.3 | **96.2**±0.1 | 63.1±1.7 | **73.9**±2.2 |
| DeepSeek-V3/Fake. | 3.4±0.7 | **22.8**±1.7 | 2.4±0.5 | **18.5**±2.0 | 80.5±0.3 | **93.6**±0.5 | 60.4±2.5 | **73.1**±2.4 |
| BERT/Large. | 3.9±0.4 | **24.5**±1.4 | 3.6±0.5 | **20.3**±1.8 | 81.7±0.2 | **91.0**±0.8 | 64.2±1.1 | **79.4**±1.1 |
| BERT/Fake. | 4.1±0.3 | **24.2**±1.7 | 2.9±0.1 | **22.6**±1.7 | 78.2±1.3 | **89.8**±0.8 | 62.2±2.5 | **76.5**±2.1 |
| YOLOv8/ImageNet | 23.5±1.5 | **31.9**±1.7 | 19.4±1.8 | **26.1**±2.1 | 78.3±3.2 | **95.2**±1.3 | 68.3±2.3 | **81.8**±2.5 |
| YOLOv8/Caltech101 | 16.2±1.3 | **23.6**±1.6 | 18.7±1.3 | **23.5**±1.5 | 84.3±0.5 | **96.7**±0.8 | 72.9±1.3 | **82.1**±2.1 |
| YOLOv8/CUB | 17.8±1.5 | **22.5**±1.1 | 20.1±1.2 | **24.1**±1.6 | 83.5±0.1 | **95.5**±0.2 | 68.8±2.7 | **75.8**±1.7 |
| ViT/ImageNet | 22.3±1.2 | **31.5**±1.4 | 20.7±1.1 | **23.9**±1.5 | 82.4±0.5 | **93.6**±0.3 | 75.2±0.9 | **84.3**±1.2 |
| ViT/Caltech101 | 18.9±1.8 | **26.7**±2.0 | 19.4±1.7 | **24.1**±1.4 | 83.5±0.8 | **97.0**±0.4 | 71.2±1.2 | **80.9**±0.7 |
| ViT/CUB | 21.9±1.0 | **29.1**±1.0 | 26.4±1.1 | **30.7**±1.0 | 85.6±0.3 | **94.3**±1.2 | 74.5±1.3 | **82.0**±1.0 |
| ResNet-50/ImageNet | 21.5±1.2 | **29.9**±1.6 | 20.1±1.4 | **29.3**±2.1 | 81.8±2.3 | **97.9**±0.5 | 76.8±2.1 | **83.6**±1.8 |
| ResNet-50/Caltech101 | 23.4±1.2 | **27.8**±1.2 | 19.7±1.3 | **28.4**±1.2 | 84.5±4.4 | **96.4**±1.0 | 77.3±1.4 | **85.1**±1.0 |
| ResNet-50/CUB | 17.0±1.1 | **26.1**±1.2 | 19.1±1.4 | **20.6**±1.3 | 78.3±4.3 | **89.5**±0.2 | 56.0±2.0 | **72.7**±3.9 |
| Qwen2.5-VL | 10.5±0.9 | **21.8**±1.9 | 8.5±1.1 | **21.8**±1.9 | 75.4±2.3 | **93.1**±0.5 | 64.7±1.2 | **73.2**±1.4 |

Table 3: Average AOPC (higher is better) and $\mathrm{accuracy_a}$ (lower is better) of LIME, KSHAP, and their ConLUX-augmented versions.

| Models | AOPC (%)↑ | | | | $\mathrm{Accuracy_a}$ (%)↓ | | | |
|---|---|---|---|---|---|---|---|---|
| | LIME | LIME* | KSHAP | KSHAP* | LIME | LIME* | KSHAP | KSHAP* |
| DeepSeek-V3/Large. | 22.1±4.1 | **46.4**±3.3 | 32.7±3.3 | **52.5**±3.3 | 77.3±1.9 | **49.3**±0.8 | 73.1±1.7 | **46.3**±1.2 |
| DeepSeek-V3/Fake. | 21.7±1.7 | **48.3**±2.1 | 30.4±1.4 | **49.6**±1.9 | 76.4±1.3 | **51.1**±1.1 | 71.6±1.4 | **49.6**±1.1 |
| BERT/Large. | 24.3±3.2 | **45.6**±3.6 | 37.9±4.1 | **55.3**±2.7 | 75.7±1.2 | **47.7**±1.9 | 60.3±1.7 | **40.2**±2.1 |
| BERT/Fake. | 25.7±2.7 | **47.2**±3.1 | 33.8±3.0 | **51.9**±3.4 | 76.2±1.3 | **53.2**±2.0 | 67.9±1.2 | **45.1**±1.5 |
| YOLOv8/ImageNet | 38.9±2.0 | **51.4**±1.8 | 41.7±1.5 | **58.7**±1.7 | 14.9±1.1 | **4.7**±1.3 | 23.9±1.3 | **6.5**±0.9 |
| YOLOv8/Caltech101 | 44.0±4.0 | **62.2**±2.8 | 43.1±3.3 | **59.6**±3.6 | 23.8±1.5 | **13.7**±1.2 | 21.1±1.6 | **11.3**±1.6 |
| YOLOv8/CUB | 51.8±3.2 | **63.7**±3.0 | 52.9±3.1 | **62.4**±3.2 | 7.0±1.2 | **1.0**±0.5 | 8.1±1.4 | **3.2**±1.2 |
| ViT/ImageNet | 45.8±2.8 | **56.3**±1.7 | 46.8±2.1 | **62.0**±3.3 | 22.4±1.2 | **5.4**±1.0 | 18.6±1.4 | **5.3**±1.4 |
| ViT/Caltech101 | 47.3±2.8 | **64.5**±3.5 | 48.1±3.0 | **67.3**±3.4 | 22.3±1.3 | **13.4**±1.2 | 21.3±1.9 | **11.8**±1.5 |
| ViT/CUB | 59.2±2.1 | **70.8**±2.1 | 62.9±2.2 | **69.7**±2.2 | 6.5±0.8 | **1.6**±0.3 | 8.2±1.0 | **3.2**±0.7 |
| ResNet-50/ImageNet | 22.3±3.0 | **32.3**±2.9 | 24.1±2.8 | **33.9**±2.4 | 21.1±1.4 | **3.6**±1.0 | 19.4±0.8 | **3.9**±0.9 |
| ResNet-50/Caltech101 | 29.7±3.7 | **48.1**±3.2 | 27.5±3.4 | **50.1**±2.8 | 17.7±1.3 | **7.9**±1.2 | 15.8±1.3 | **6.3**±1.0 |
| ResNet-50/CUB | 46.5±3.0 | **56.3**±3.5 | 45.2±3.4 | **56.7**±3.0 | 4.9±1.1 | **2.1**±0.9 | 7.4±1.6 | **4.5**±1.0 |
| Qwen2.5-VL | 24.5±2.1 | **40.7**±2.4 | 31.6±2.8 | **39.8**±2.7 | 21.2±1.2 | **5.1**±0.9 | 16.9±1.4 | **3.0**±0.6 |

bution $\mathbb{D}_{\boldsymbol{x}}$, their corresponding explanation $g$, and a performance metric $L$ (e.g. accuracy, F1 score, etc.), we defined the (in-)fidelity as $E_{\boldsymbol{z}\sim\mathbb{D}_{\boldsymbol{x}}}[L(\mathbf{1}_{\{\boldsymbol{z}|f(\boldsymbol{z})=f(\boldsymbol{x})\}}(\boldsymbol{z}), g(\boldsymbol{z}))]$, which indicates the performance of using $g$ to approximate the local behavior of $f$, i.e., predict if f($\boldsymbol{z}$) is the same as $f(\boldsymbol{x})$. Here, we used accuracy as the performance metric.

For a fair comparison, we calculate fidelity in different neighborhoods for each setup: 1) When comparing ConLUX-augmented explanations with their feature-level versions, we use feature-level neighborhood. 2) When comparing ConLUX unified explanations with other concept-based explanations, we use concept-level neighborhood.

### 4.2.3 RESULTS

Table 2 and 3 show the fidelity of Anchors, LORE, LIME, KSHAP, and their ConLUX-augmented versions. For Anchors and LORE, ConLUX improves the average coverage by 11.2% and 9.5%, and the average precision by 13.0% and 10.6%, respectively. For LIME and KSHAP, ConLUX improves the average AOPC by 0.164 and 0.151, and decreases the average $\mathrm{accuracy_a}$ by 14.8% and 13.8%, respectively. We did paired t-tests for setups that only differ in whether to apply ConLUX, which indicates with over 99% confidence the improvement is significant.

Table 4 shows the fidelity of ConLUX-augmented explanations and state-of-the-art concept-based explanation methods: TBM and LACOAT for text tasks, and EAC and ConceptLIME for image tasks, which shows that ConLUX helps KSHAP to achieve higher fidelity than these methods on all tasks, and ConLUX local unified explanations further achieve 4.52% more fidelity than ConLUX-augmented KSHAP explanations.

Table 4: Average accuracy (%) (higher accuracy is better) of TBM, EAC, ConLUX-augmented KSHAP (denoted as KSHAP*), and ConLUX unified explanations.

| Models | TBM | LACOAT | EAC | ConceptLIME | KSHAP* | ConLUX |
|---|---|---|---|---|---|---|
| DeepSeek-V3/Large. | 78.3 | 65.8 | – | – | 87.1 | **90.7** |
| DeepSeek-V3/Fake. | 75.4 | 66.7 | – | – | 82.3 | **87.3** |
| BERT/Large. | 82.4 | 73.4 | – | – | 86.3 | **92.6** |
| BERT/Fake. | 81.4 | 75.2 | – | – | 85.6 | **91.8** |
| YOLOv8/ImageNet | – | – | 82.7 | 81.3 | 88.1 | **92.4** |
| YOLOv8/Caltech101 | – | – | 84.6 | 83.9 | 88.7 | **92.1** |
| YOLOv8/CUB | – | – | 76.8 | 72.6 | 86.7 | **89.3** |
| ViT/ImageNet | – | – | 83.4 | 79.3 | 88.6 | **91.6** |
| ViT/Caltech101 | – | – | 82.7 | 84.1 | 89.4 | **92.6** |
| ViT/CUB | – | – | 74.3 | 72.1 | 86.4 | **90.3** |
| ResNet-50/ImageNet | – | – | 78.1 | 77.9 | 85.0 | **90.2** |
| ResNet-50/Caltech101 | – | – | 79.7 | 76.8 | 85.4 | **91.5** |
| ResNet-50/CUB | – | – | 77.4 | 77.1 | 83.1 | **89.1** |

Table 5: Human Evaluation Results on EAC, ConLUX-augmented Anchors (denoted as Anchors*), and ConLUX-augmented LORE (denoted as LORE*).

| Methods | Sufficient Conditions | | Counterfactuals | |
|---|---|---|---|---|
| | EAC | Anchors* | EAC | LORE* |
| coverage$_u$ (%) | 57.0 | 60.0 | 29.3 | 36.1 |
| Precision$_u$ (%) | 63.9 | 72.0 | 82.8 | 97.0 |

## 4.3 HUMAN EVALUATION

ConLUX provides explanations in various forms beyond attributions, including sufficient conditions and counterfactuals, which can more effectively support users in downstream decision-making tasks.

Specifically, sufficient conditions help users anticipate the model's behavior on unseen inputs, while counterfactuals help understand how the model's prediction changes under specific input modifications. To evaluate the usefulness of introducing these explanation forms, we conducted a user study following the setup in Ribeiro et al. (2018); Warren et al. (2022); Hase & Bansal (2020), comparing our concept-based sufficient conditions and counterfactuals with concept-based attribution explanations provided by EAC (Sun et al., 2023) across two tasks.

We recruited 18 subjects, all graduate students with machine learning experience but no explainable AI expertise. Each subject completed a questionnaire with 10 tests per task.

For the sufficient condition task, in each test, participants were first shown an image randomly sampled from the ImageNet dataset, along with YOLOv8's prediction and an explanation randomly selected from EAC or ConLUX-augmented Anchors explanations. They were then shown 5 new images generated by concept-level perturbing the original image, and asked to predict whether the model would output the same label for each new image based on the explanation. They could answer "yes" or "no." We evaluate responses using two metrics: coverage$_u$, the proportion of "yes" responses, and precision$_u$, the proportion of correct predictions among the "yes" responses.

The counterfactual task follows a similar setup, except that participants were shown ConLUX-augmented LORE explanations instead of Anchors and were asked to predict whether the model would output a *different* label for each new image.

Table 5 shows the average coverage$_u$ and precision$_u$ of the two tasks. The results show that ConLUX-augmented explanations outperform EAC by 3.0% in coverage$_u$ and 8.1% in precision$_u$ for the sufficient condition task, and by 6.8% and 14.2% respectively for the counterfactual task. These findings indicate that ConLUX-augmented explanations are more effective in helping users apply explanations to reason about model behavior.

## 4.4 RUNTIME OVERHEAD

We show the execution time of the explanation fidelity experiments on image data in Figure 4. As shown in Figure 4(a), ConLUX does require additional computation time due to generative model calls. However, the running time of ConLUX-augmented methods is practically acceptable.

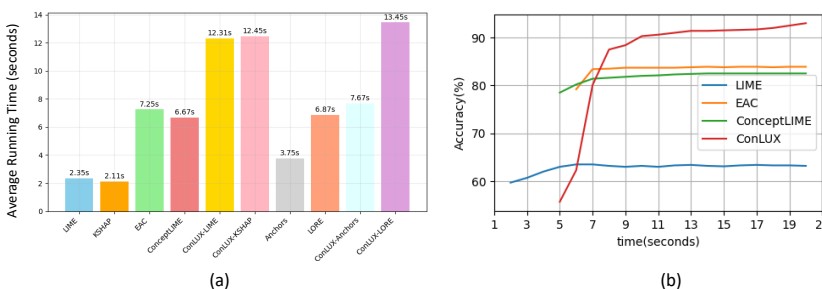

Figure 4: (a) Average running time of explanation methods we used on image data. (b) Matched-budget comparisons between LIME, EAC, ConceptLIME and ConLUX local unified explanations.

We also conducted matched-budget comparisons (measured in running time) to evaluate the utility of each method under the same computational constraints. We followed the experimental settings in Section 4.2.1, explaining ViT on the ImageNet dataset with LIME, EAC, and ConceptLIME, and ConLUX local unified explanations. Figure 4(b) shows the results. While ConLUX may start with lower fidelity at very low budgets due to the overhead of generative model calls, it surpasses baseline methods as the budget increases. This indicates that when users have sufficient computational time, ConLUX can effectively utilize it to provide higher-fidelity explanations. Especially when ConLUX surpasses baseline methods, the absolute time cost is still not high.

### 4.5 ROBUSTNESS TO CONCEPT-FEATURE MAPPING MODELS

As ConLUX relies on generative models for concept–feature mapping, we evaluate its robustness under different generative model choices. We replace the generative model used in ConLUX with alternative ones and measure the resulting perturbation fidelity. Specifically, we test two additional Qwen2.5 models for text perturbations and Latent Consistency Model (Luo et al., 2023) for image perturbations. Table 1 summarizes the results, showing that ConLUX maintains high perturbation fidelity across all tested models.

This demonstrates that ConLUX is flexible in choosing the generative model for concept–feature mapping, which brings two benefits: (1) users can select models based on their own preferences or resource constraints, and (2) users can employ multiple generative models to increase the diversity of generated samples and reduce potential biases from relying on a single model.

Further experimental settings and results are provided in Appendix C.

## 5 CONCLUSION

We have proposed ConLUX, a general framework that elevates local model-agnostic explanation methods to the concept level. ConLUX offers unified explanations combining attributions, sufficient conditions, and counterfactuals. This satisfies diverse user needs and fills the current gap of concept-based explanations lacking forms beyond attributions. ConLUX achieves this by utilizing large pre-trained models to extract high-level concepts, and extending perturbation models to sample in the concept space. We have instantiated ConLUX on Anchors, LIME, LORE, and Kernel SHAP, and demonstrated the state-of-the-art performance of ConLUX by empirical evaluations. ConLUX shows that it is unnecessary to design concept-based explanation methods from scratch, as existing local model-agnostic methods can be easily elevated to concept level in a lightweight manner.

## ETHICS STATEMENT

This work complies with the ICLR Code of Ethics . We have carefully considered potential broader impacts, including possible risks and benefits to society, and have conducted our research in accordance with principles of scientific integrity, fairness, and responsible stewardship.

The participants of our user study will not incur potential risk, and we have told them the description and instructions of our study before they join. Our study is approved by the IRB of our school. We have paid the participants as double the minimum wage of our city. The participants are free to leave at any time during the study. We have also told them that their data will be used for research purposes only and will not be shared with any third parties.

## REPRODUCIBILITY STATEMENT

The code of our framework and experiments are available at `https://anonymous.4open.science/r/ConLUX/`.

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

# A  THE USE OF LARGE LANGUAGE MODELS

We use LLMs to refine and polish human writing, and find related work with DeepResearch. We do not use LLMs to generate the main content or ideas of this paper.

# B  THE CONLUX FRAMEWORK (CONTINUED)

In this section, we introduce the details of incorporating ConLUX into existing local explanation methods.

We first follow Section 3 to introduce how we extend each part for text models in detail.

## B.1  PRODUCING CONCEPT

ConLUX provides predicates that describe high-level concepts **following existing concept-based methods**.

Specifically, we follow Ludan et al. (2023) for text data and Sun et al. (2023) for image data, and keep all theirs settings.

## B.2  CONCEPT-FEATURE MAPPING

**Text Data.** We the the following prompt for predicate-feature mapping:

> Generate a sentence similar to a given sentence from the domain of {} dataset. The dataset's description is that {}.
>
> The generated sentence satisfies given concepts. Before generating the sentence, carefully read the description of each concept to understand the properties that the generated sentence must satisfy, think about how the sentence satisfies the concepts first, and then create the sentence that satisfies the concepts.
>
> Format your response as a JSON with string keys and string values. Below is an example of a valid JSON response. The JSON contains keys `thoughts`, and `answer`. End your response with \###
>
> ---
>
> Concepts:
> 1. Concept 1
> 2. Concept 2
> ...
>
> Response JSON:
> {
>   "thoughts": "In this section, you explain which snippets in your text support the concepts. Be as objective as possible and ignore irrelevant information. Focus only on the snippets and avoid making guesses.",
>   "answer": "A sentence that satisfies the concepts."
> }
> ###
>
> Two examples of this task being performed can be seen below. Note that the answer should be in 5 to 20 words and should be a single sentence.
>
> Example 1:
>
> Concepts:
> 1. The plot of the text is exciting, captivating, or engrossing. It may have

Table 6: Explanation fidelity (accuracy) of using ConLUX with different concept-feature mapping models.

| Model | Accuracy |
|---|---|
| **DeepSeekV3** | 92.6 |
| Qwen2.5 72B | 91.8 |
| Qwen2.5 7B | 90.3 |
| **Blended Latent Diffusion** | 91.6 |
| Latent Consistency Model | 91.4 |

unexpected twists, compelling conflicts, or keep the reader eagerly turning pages.
2. The characters in the movie are portrayed in a realistic and convincing manner. Their actions, dialogue, emotions, motivations, and development feel authentic and relatable, making them believable to the audience.
3. The narrative structure of the text is confusing or unclear, making it difficult to follow or comprehend the events happening within the story.
4. The text introduces some original elements or takes minor risks in the plot development, but overall, it lacks a truly unique or innovative narrative.

Response JSON:
{
   "thoughts": "The snippet 'the silly and crude storyline' mentions a storyline that is described as silly and crude, indicating a lack of creativity and reliance on clichéd plot devices. The phrase 'real issues tucked between the silly and crude storyline' suggests a potentially confusing structure. It also implies real conflicts, satisfying the exciting plot concept. The term 'real issues' also supports the realistic character portrayal.",
   "answer": "it's about issues most adults have to face in marriage and i think that's what i liked about it – the real issues tucked between the silly and crude storyline."
}
###

Example 2:
...
###

Perform the task below, keeping in mind to limit the response to 5 to 20 words and a single sentence. Return a valid JSON response ending with ###

Concepts:
{}

Response JSON:

**Image Data.** We use the following prompt for predicate-feature mapping:

Integrate this area into the background of the image.

## C EMPIRICAL EVALUATION (CONTINUED)

### C.1 ROBUSTNESS TO CONCEPT-FEATURE MAPPING MODELS (CONTINUED)

Besides the results in Section 4.5, we additionally conducted experiments to show the explanation fidelity of using different generative models for concept-feature mapping. We explain a BERT model on the Large Movie Review dataset, and a ViT modle on ImageNet using ConLUX local unified

Table 7: Accuracy of using different prompts to generate perturbations for text data.

| Prompt | CoNLL | Onto. | Large. |
|--------|-------|-------|--------|
| Original | 97.5 | 98.1 | 97.7 |
| Paraphrased | 97.8 | 98.0 | 97.9 |
| Shortened | 96.2 | 96.8 | 96.5 |

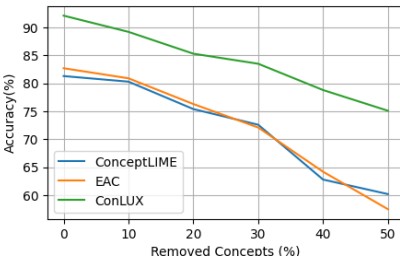

Figure 5: Effect of concept absence on explanation fidelity.

explanation. Table 6 shows the results, which futher support that ConLUX is robust to different generative model choices.

## C.2   ROBUSTNESS TO PROMPTS

We also conducted an experiment using different prompts to generate perturbations. We applied two other settings, one to ask GPT-4o to make our prompt template to at only 70% of the original perturbation, and the other to ask GPT-4o to generate a prompt that is similar to the original prompt, but paraphrased. The results are shown in Table 7. We can see that the perturbation model is robust to different prompts.

## C.3   ROBUSTNESS TO CONCEPTS

To show the robustness of ConLUX to when concepts are not ideal, we conducted two ablation studies to investigate the effects of concept absence and concept correlation on explanation fidelity using ImageNet.

### C.3.1   CONCEPT ABSENCE

When explaining YOLOv8 on ImageNet, we randomly removed a certain percentage of concepts from the concept set used for generating explanations. Figure 5 shows how explanation fidelity changes as we vary the percentage of removed concepts. The results show that ConLUX exhibits greater robustness to concept removal compared to baseline concept-based explanation methods.

### C.3.2   CONCEPT CORRELATION

We conducted experiments on text data, specifically explaining BERT on the Large Movie Review dataset. In these experiments, we added a correlated concept to the concept set and measured the explanation fidelity both before and after adding the correlated concept. The results show that the accuracy slightly decreases from 92.6% to 91.9%, indicating that ConLUX is robust to concept correlation.

## C.4   FLEXIBILITY OF SUPPORTING VARIOUS CONCEPTS

ConLUX is a flexible framework to generate explanations with various concepts. Besides those we have shown in Section 4, we additionally implmented ConLUX for two more types of concepts to demonstrate its flexibility.

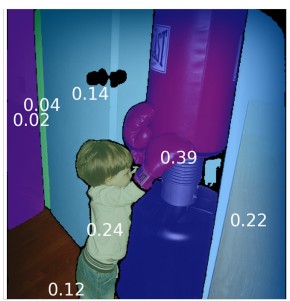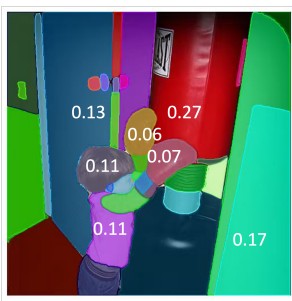

Figure 6: ConLUX-LIME explanation with hierarchical concepts.

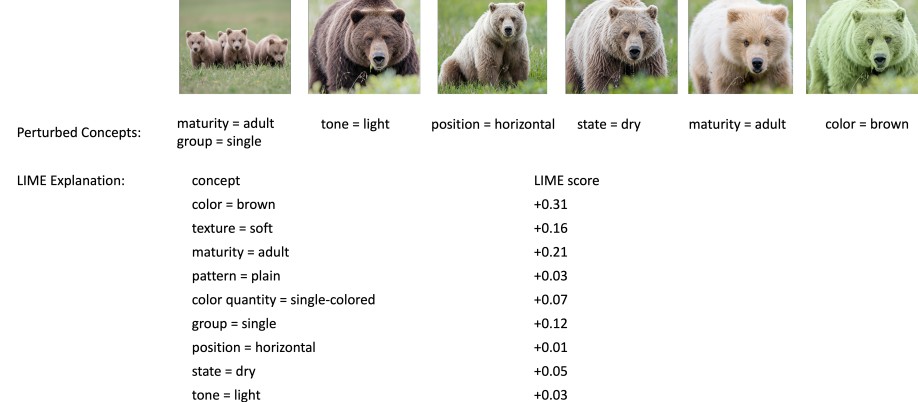

| Perturbed Concepts: | maturity = adult group = single | tone = light | position = horizontal | state = dry | maturity = adult | color = brown |
|---|---|---|---|---|---|---|

| LIME Explanation: | concept | LIME score |
|---|---|---|
| | color = brown | +0.31 |
| | texture = soft | +0.16 |
| | maturity = adult | +0.21 |
| | pattern = plain | +0.03 |
| | color quantity = single-colored | +0.07 |
| | group = single | +0.12 |
| | position = horizontal | +0.01 |
| | state = dry | +0.05 |
| | tone = light | +0.03 |

Figure 7: ConLUX-LIME perturbation samples with attributes as concepts and the corresponding explanation.

### C.4.1 HIERARCHICAL CONCEPTS

HIPIE Wang et al. (2023) can extract hierarchical concepts, which can then be directly used in ConLUX to produce explanations. We have implemented this, and Figure 6 shows an example of hierarchical ConLUX-LIME explanations.

### C.4.2 ATTRIBUTES AS CONCEPTS

The Open-Vocabulary Attribute Detection dataset Bravo et al. (2023) provides attribute-level annotations (e.g., color, texture) for objects in COCO dataset. This allows ConLUX to generate explanations based on these object attributes instead of replacing objects entirely. Figure 7 shows some perturbation samples generated by ConLUX-LIME using attributes as concepts and the corresponding explanation.

### C.5 RUNTIME OVERHEAD (CONTINUED)

We have discussed the computational cost of ConLUX in Section 4.4, we also provide the moneytary cost and counts the calls of the generative models in Table 8. As shown, ConLUX does require additional computation time due to generative model calls. However, we note:

- **Zero monetary cost:** Without compromising performance, all our experiments (Section 4 and Table 6) can use LLMs that can run on a single GPU without relying on commercial APIs, resulting in negligible monetary cost.

- **Acceptable overhead:** The running time of ConLUX-augmented methods is practically acceptable.

Table 8: Average runtime overhead and monetary cost of different explanation methods on image data.

| Method | Avg Perturbations | Avg LM Calls | Avg Time (s) | Avg Cost ($) |
|---|---|---|---|---|
| LIME | 1000 | 0 | 2.35 | 0 |
| ConLUX-LIME | 1000 | 1000 | 12.31 | 0 |
| EAC | 1000 | 0 | 7.25 | 0 |
| ConceptLIME | 1000 | 1000 | 6.67 | 0 |
| KSHAP | 1000 | 1000 | 2.11 | 0 |
| ConLUX-KSHAP | 1000 | 1000 | 12.45 | 0 |
| Anchors | 193 | 0 | 3.75 | 0 |
| ConLUX-Anchors | 71 | 71 | 7.67 | 0 |
| LORE | 851 | 0 | 6.87 | 0 |
| ConLUX-LORE | 873 | 873 | 13.45 | 0 |

## C.6 EXPLANATION FIDELITY EVALUATION DETAILS

### C.6.1 SETUP DETAILS

We experimented on two machines, one with an Intel i9-13900K CPU, 128 GiB RAM, and RTX 4090 GPU, and another with Intel(R) Xeon(R) Silver 4314 CPU, 256GiB RAM, and 4 RTX 4090 GPUs.

To measure the fidelity improvement brought by ConLUX, we keep all hyperparameters the same for both vanilla and augmented methods.

For LIME and KSHAP, we set the number of sampled inputs to 1000 except for explaining DeepSeekV3.

For Anchors, we follow the default settings.

For LORE, we set $ngen = 5$.

For the DeepSeek-V3 model, when applying it to the movie review sentiment analysis task, we simply use the following prompt:

> From now on, you should act as a sentiment analysis neural network. You should classify the sentiment of a movie review as positive or negative. If the sentence is positive, you should reply 1. Otherwise, if it's negative, you should reply 0. There may be some words that are masked in the sentence, which are represented by <UNK>. The input sentence may be empty, which is represented by <EMPTY>. You will be given the sentences to be classified, and you should reply with the sentiment of the sentence by 1 or 0.
> There are two examples:
> Sentence:
> {sentence 1 from training set}
> Sentiment:
> {label 1}
> Sentence:
> {sentence 2 from training set}
> Sentiment:
> {label 2}
> You must follow this format. Then I'll give you the sentence. Remember Your reply should be only 1 or 0. Do not contain any other content in your response. The input sentence may be empty.
> Sentence:
> {The given sentence}
> Sentiment:

Table 9: Average accuracy (%) (higher accuracy is better) of TBM, EAC, ConLUX-augmented KSHAP (denoted as KSHAP*), and ConLUX unified explanations.

| Models | AOPC↑ | | | | | Accuracy$_a$(↓) | | | | |
|---|---|---|---|---|---|---|---|---|---|---|
| | TBM | LACOAT | EAC | ConceptLIME | KSHAP* | TBM | LACOAT | EAC | ConceptLIME | KSHAP* |
| DeepSeek-V3/Large. | 47.8 | 46.3 | – | – | **54.3** | 49.6 | 49.8 | – | – | **44.1** |
| DeepSeek-V3/Fake. | 44.2 | 45.2 | – | – | **51.4** | 47.4 | 48.6 | – | – | **45.7** |
| BERT/Large. | 51.0 | 49.6 | – | – | **57.7** | 46.5 | 43.1 | – | – | **38.2** |
| BERT/Fake. | 47.4 | 47.8 | – | – | **53.1** | 49.1 | 45.9 | – | – | **39.3** |
| YOLOv8/ImageNet | – | – | 52.9 | 51.1 | **62.1** | – | – | 6.2 | 5.9 | **5.3** |
| YOLOv8/Caltech101 | – | – | 54.9 | 51.6 | **62.0** | – | – | 8.3 | 8.4 | **7.2** |
| YOLOv8/CUB | – | – | 52.4 | 51.5 | **60.5** | – | – | 8.9 | 9.1 | **7.8** |
| ViT/ImageNet | – | – | 57.1 | 54.8 | **65.4** | – | – | 7.1 | 6.4 | **4.5** |
| ViT/Caltech101 | – | – | 62.9 | 59.4 | **69.3** | – | – | 11.5 | 11.9 | **10.3** |
| ViT/CUB | – | – | 54.1 | 56.8 | **64.7** | – | – | 6.3 | 6.1 | **5.1** |
| ResNet-50/ImageNet | – | – | 22.1 | 25.3 | **34.9** | – | – | 4.6 | 4.7 | **2.9** |
| ResNet-50/Caltech101 | – | – | 41.4 | 43.6 | **53.4** | – | – | 7.4 | 7.8 | **5.2** |
| ResNet-50/CUB | – | – | 32.6 | 30.7 | **39.1** | – | – | 5.2 | 6.0 | **4.1** |

When using Deepseek-V3 for fake news detection, we apply similar prompts to those used in other tasks. We obtain the classification probabilities by applying a softmax function to the raw probabilities of returning the two tokens: 0 and 1.

For Qwen2.5-VL, we directly prompt it to answer questions in VQAv2, and obtain the classification probabilities by applying a softmax function to the raw probabilities in Yes/No questions.

### C.6.2  RESULTS (CONTINUED)

In this section, we present the fidelity results that are not present in the main paper limited by space.

As Table 4 shows, either ConLUX-augmented KSHAP or ConLUX unified explanation can outperform TBM and EAC, two state-of-the-art concept-based methods, in terms of fidelity. We additionally provide the results of deletion experiments on TBM, EAC and ConLUX-augmented KSHAP in Table 9.

### C.7  HUMAN EVALUATION

We introduce more detail about human evaluation in this section.

**User Screening.** We recruited the users from students of Computer Science school in our university, and screened them before filling out the questionnaires. Therefore, the set is relatively small. To make sure our users understood the meaning of the explanation and would predict the model output based on the explanation rather than their own feelings, we made up a test with a counter-intuitive explanation. Then we let our users predict the model outputs of ten perturbed images. We remain the users that follow the counter-intuitive explanation.

**Questionnaires details.** The questionnaires are similar for all users with minor variations in the order of presentation. We presented the ten questions, Q1, Q2,..., and Q10 in a random order. For each question, we presented one of the explanations generated by EAC and ConLUX. For example, for the first question, user A received an explanation generated by EAC, while user B may received an explanation generated by ConLUX. The users were asked to predict the model output based on the explanation. Also, we presented the five images to be predicted in a random order. Figure 8, 9, 10, and 11 show example test questions in the questionnaire.

Input

Output: space_shuttle

Explanation

If the highlighted area **remains**, the model will classify the image as **space_shuttle.**

According to the explanation, please predict if the following image will be classified as **space_shuttle**
- If you think the image should be classified as **space_shuttle**, please write "Y"
- If you think the image should be not classified **space_shuttle**, or you cannot make decision according to the explanation, please write "N"

1. (   )    2. (   )    3. (   )    4. (   )    5. (   )

Figure 8: An question in the task of comparing EAC and ConLUX-augmented Anchors. This question provides user an ConLUX-augmented Anchors explanation.

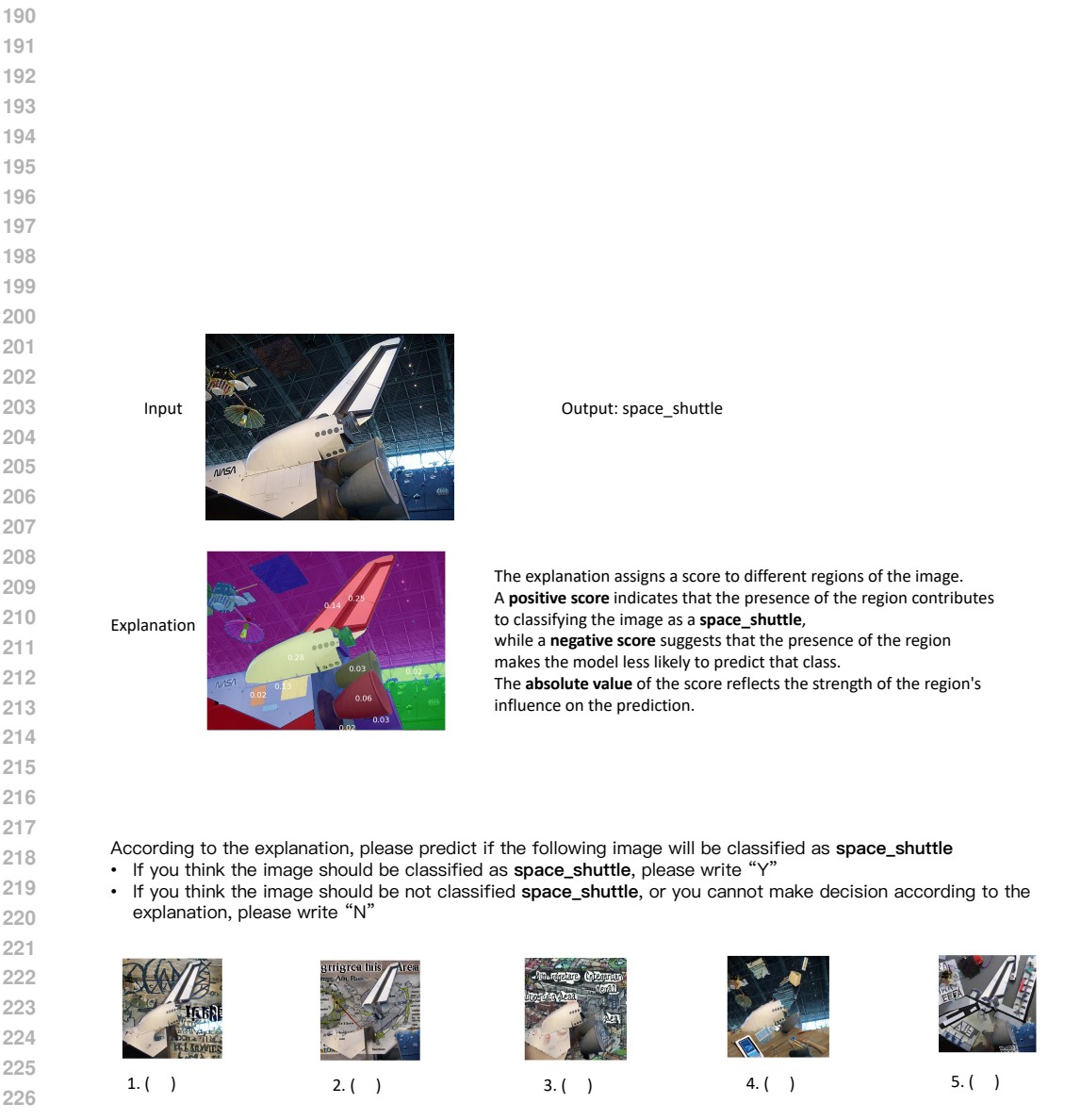

Figure 9: A question in the task of comparing EAC and ConLUX-augmented Anchors. This question provides the user with an EAC explanation.

Input          Output: Stage

Explanation    If the highlighted area **changes**, the model will classify the image as **another class.**

According to the explanation, please predict if the following image will be classified **another class**.
- If you think the image should be classified as **another class**, please write "Y"
- If you think the image should be not classified **another class**, or you cannot make decision according to the explanation, please write "N"

1. (   )          2. (   )          3. (   )          4. (   )          5. (   )

Figure 10: A question in the task of comparing EAC and ConLUX-augmented LORE. This question provides the user a ConLUX-augmented LORE explanation.

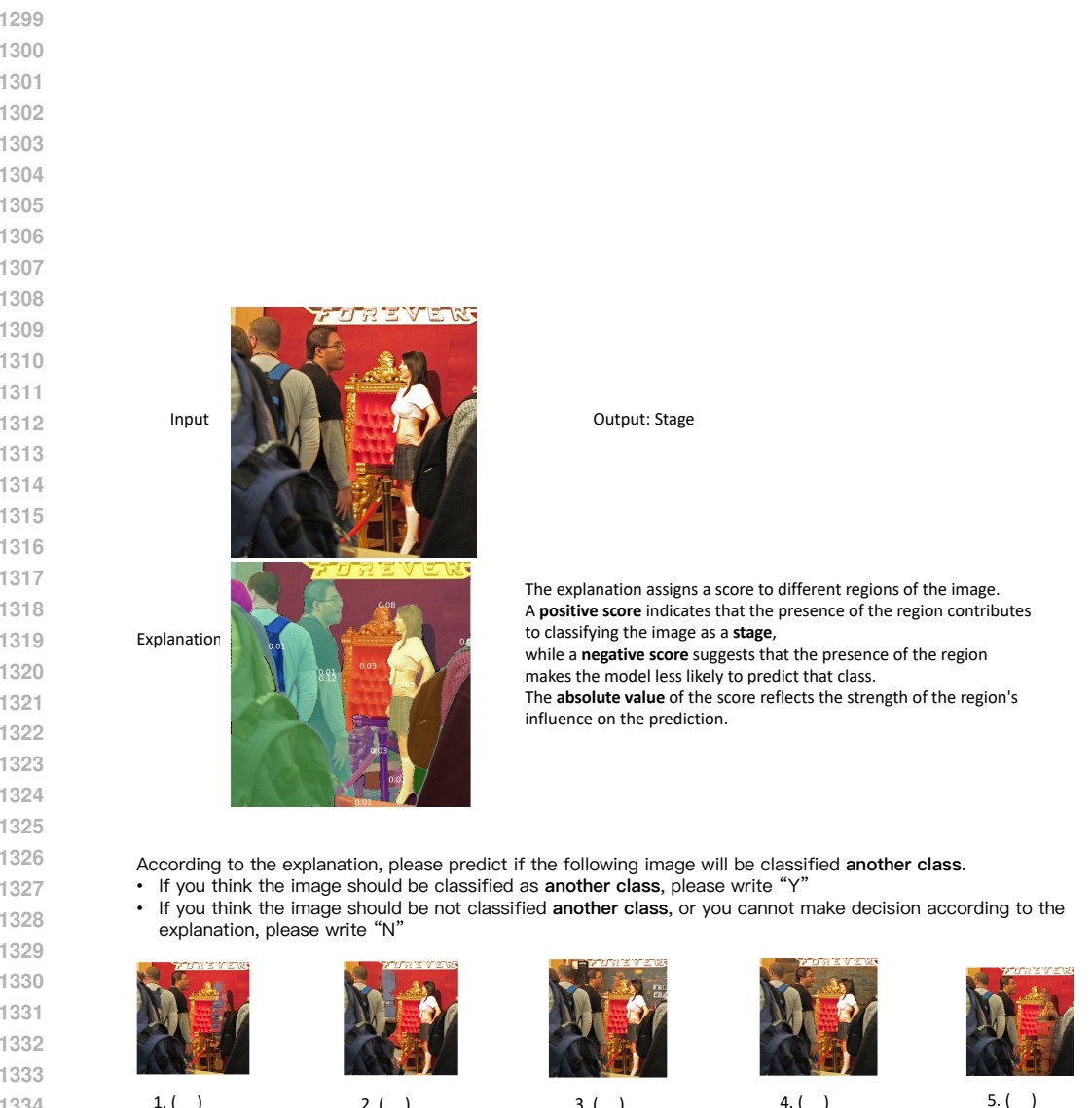

Figure 11: An question in the task of comparing EAC and ConLUX-augmented LORE. This question provides user an EAC explanation.

