# OpenReview forum: "Concept-Based Local Unified Explanations"
_ICLR.cc/2026/Conference — Submitted to ICLR 2026_

### Official Review · Reviewer_aiQU · 2025-10-29

**Soundness:** 2
**Presentation:** 2
**Contribution:** 2
**Rating:** 4
**Confidence:** 3

**Summary:**

This paper addresses the limitations of current concept-based model-agnostic explanation methods, which mainly focus on attribution tasks and offer limited types of explanations. The authors propose ConLUX, a general and lightweight framework that systematically extends local model-agnostic explanation techniques—such as LIME, Kernel SHAP, Anchors, and LORE—into the concept-level domain. By leveraging large pre-trained models to perform concept-level perturbations, ConLUX enables the generation of unified concept-based explanations, including attributions, sufficient conditions, and counterfactuals.

**Strengths:**

The paper makes a contribution by bridging model‑agnostic and concept‑based explanations, positioning ConLUX as a framework capable of generalizing across existing local explanation tools. Beyond the conceptual advancement, the empirical work is comprehensive: ConLUX is instantiated on four popular explanation methods (LIME, Kernel SHAP, Anchors, and LORE) and consistently improves their fidelity metrics, demonstrating broad applicability and robustness. The approach also exhibits comparative superiority over current concept‑level explanation methods across both text and image tasks, achieving higher fidelity than TBM, LACOAT, EAC, and ConceptLIME.

**Weaknesses:**

In my view, the primary concern with this paper lies in its motivation. The introduction claims that existing methods "lack support for richer explanation types such as sufficient conditions and counterfactuals." However, the paper does not sufficiently explain why these types of explanations are important or what practical or theoretical gap they fill. In other words, the “so what” question remains unanswered. Although the authors define notions such as sufficient conditions and counterfactuals later in the paper (Lines 169–182), there is limited discussion or justification about their significance. As a result, it feels as though these concepts were added to extend the existing XAI pipeline in a more mechanical way, rather than being driven by a strong underlying motivation or clear end-user need.

Second, the authors state that their tool “elevates existing local model-agnostic explanation methods to the concept level with minimal user effort.” This raises the question of what kind of contribution the paper is positioning itself as. Should we interpret it primarily as a tool-based or implementation-level contribution? From my understanding, there are already several mature toolkits that aim to unify and streamline explanation methods—for example, the Captum API. It is not entirely clear what novel capability ConLUX introduces beyond what such existing frameworks already offer.

Finally, a more minor but related concern: the paper emphasizes that the proposed method works with “minimal user effort,” yet the framework relies on large pre-trained models for concept-level perturbations. Given that LLMs often require substantial computational resources—sometimes with billions of parameters—it is difficult to reconcile this reliance with the claim of minimal user effort. Some clarification on what “minimal effort” precisely means in this context would help readers better understand the practical usability of the approach.

**Questions:**

**Q1. Motivation and Significance**
- Could you elaborate on **why providing sufficient conditions and counterfactual explanations** is practically or theoretically important?
- For instance, are there concrete end-user needs, application domains, or specific decision-making settings where the lack of such explanations has created known limitations?
- If possible, please include empirical or user-study evidence, or a literature gap analysis, that demonstrates the necessity of supporting these richer explanation types.

**Q2. Positioning and Novelty of the Contribution**
- How should readers understand your contribution in relation to existing XAI toolkits (e.g., Captum or other integrated explanation frameworks)?
- It would help to explicitly state whether the novelty lies primarily in:
  1. a new underlying algorithmic capability,
  2. a conceptual framework for organizing explanation types, or
  3. engineering integration and usability improvements.
- A comparative table or ablation study against representative existing tools could clarify the incremental novelty or unique advantages of ConLUX.

**Q3. Definition of “Minimal User Effort”**
- Can you specify in what sense the approach requires “minimal user effort”? For example:
  - Does it refer to minimal annotation or data engineering?
  - Does it concern ease of API integration or conceptual simplicity?
- Given that your framework depends on large pre-trained models, it would be useful to qualify the meaning of “minimal effort” in light of potential computational costs.
- If possible, a brief discussion or quantitative measure (e.g., hours of setup, lines of code, computational resources) would help readers gauge the practical burden.

---

> ### Author Response · Authors · 2025-11-22
> **Response by the Authors (Part 1/2)**
>
> Thank you for your careful review and constructive comments. We address your questions below in detail:
>
> **[Q1] Motivation and Significance**
>
> As we stated in section 2, existing studies have shown that different types of explanations serve different user needs:
> - Attributions assist users in evaluating the trustworthiness of the mode [1].
> - Sufficient conditions help users anticipate the model’s behavior on unseen input [2]
> - Counterfactuals provide actionable insights for decision-making and outcome reversal [3].
>
> Specifically, when users need to predict model behavior or intervene to change model predictions, **sufficient-condition** and **counterfactual** explanations offer direct support for these tasks.
>
> Additionally, in Section 4.3 of our paper, we present a user study that evaluates the effectiveness of these explanation types. Our results show that users perform significantly better when provided with **sufficient-condition** and **counterfactual** explanations, as compared to attributions. This empirical evidence highlights the practical significance of these explanation types for real-world applications.
>
> [1] Ribeiro, Marco Tulio, et al. “‘ Why Should i Trust You?’ Explaining the Predictions of Any Classifier.” Proceedings of the 22nd ACM SIGKDD International Conference on Knowledge Discovery and Data Mining, 2016, pp. 1135–44. Google Scholar.
> [2] Ribeiro, Marco Tulio, et al. “Anchors: High-Precision Model-Agnostic Explanations.” Proceedings of the AAAI Conference on Artificial Intelligence, vol. 32, no. 1, 2018. Google Scholar.
> [3] Guidotti, Riccardo, et al. “Local Rule-Based Explanations of Black Box Decision Systems.” arXiv:1805.10820, arXiv, 28 May 2018. arXiv.org, http://arxiv.org/abs/1805.10820.
>
> **[Q2] Positioning and Novelty of the Contribution**
>
> > Compare with Captum
>
> ConLUX and Captum are orthogonal to each other. There may be a misunderstanding stemming from they both being called as "frameworks." To clarify, we ‘d like refer to **Captum** as a **"library"** and **ConLUX** as a **"method"** here.
>
> Captum is a library that consolidates various existing explanation techniques for PyTorch models, providing users with a unified interface to apply these methods. However, Captum does not introduce new explanation techniques.
>
> In contrast, **ConLUX** introduces a **novel method** that extends various existing feature-level explanation techniques, including some included in Captum, to concept-level.
>
> >  Explicitly state where the novelty lies primarily in
>
>
> The primary novelty of ConLUX lies in its **underlying algorithmic capability**. Specifically, we address a key limitation in existing concept-level explanation methods, which have been restricted to attribution-based explanations. We observe that feature-level model-agnostic explanation methods can provide diverse explanation types, but cannot generate concept-level explanations.
> To bridge this gap, obtaining concept-level explanations of various types, we propose ConLUX, a method to elevate feature-level explanation methods.
>
> > A comparative table or ablation study against representative existing tools could clarify the incremental novelty or unique advantages of ConLUX.
>
> We have conducted experiments comparing ConLUX-augmented methods with state-of-the-art concept-based explanation methods in Section 4.
> Could you please clarify which specific tools or methods you would like us to include in the comparison? We are happy to conduct additional experiments if needed.

---

> ### Author Response · Authors · 2025-11-22
> **Response by the Authors (Part 2/2)**
>
> **[Q3] Definition of “Minimal User Effort**
> > Can you specify in what sense the approach requires “minimal user effort”?
>
> The term **"minimal user effort"** in our framework refers to **conceptual simplicity** and **minimal data annotation**:
> - **Conceptual Simplicity**: ConLUX allows users to elevate existing feature level model-agnostic explanation methods to concept level without needing to understand the underlying technical details of each method.
> - **Minimal Data Annotation**: ConLUX requires no additional data annotation when applied to new datasets, making it highly accessible for users without the need for extensive manual labeling.
>
> > It would be useful to qualify the meaning of “minimal effort” in light of potential computational costs.
>
> We have conducted a cost analysis about running time on image data. The results are shown below:
>
> | Method         | Avg Time (s) |
> | -------------- | ------------ |
> | LIME           | 2.35         |
> | ConLUX-LIME    | 12.31        |
> | KSHAP          | 2.11         |
> | ConLUX-KSHAP   | 12.45        |
> | Anchors        | 3.75         |
> | ConLUX-Anchors | 7.67         |
> | Lore           | 6.87         |
> | ConLUX-Lore    | 13.45        |
>
>
> ConLUX does introduce some computational overhead compared to vanilla methods due to the additional generative model calls. However, the cost is still acceptable for most practical applications, and compared to manual concept annotation, the computational cost is minimal.
>
> > If possible, a brief discussion or quantitative measure (e.g., hours of setup, lines of code, computational resources) would help readers gauge the practical burden.
>
> As we have provided our code and prompt templates, users can easily apply ConLUX to their own datasets without writing additional code. The main time cost comes from reading the instructions and filling in the information in the prompt templates, which usually takes less than 30 minutes.
> For a user who has already used ConLUX, applying it to a new dataset requires only less than 10 minutes.
>
> As we have shown that we can use Large models that can run on a single consumer-grade GPU to complete the entire process (Section 4 and Table 6), we don't need additional computational resources when explaining target models on a local machine.
>
> We hope our response addresses your concerns.

---

> > ### Comment · Reviewer_aiQU · 2025-11-27
> >
> > Thank you for the detailed clarifications, particularly on Q1. I have updated my scores accordingly.

---

### Official Review · Reviewer_LBfo · 2025-10-30

**Soundness:** 2
**Presentation:** 3
**Contribution:** 2
**Rating:** 4
**Confidence:** 3

**Summary:**

The paper introduces ConLUX, a general framework that extends feature-level, model-agnostic local explainers such as LIME, Kernel SHAP, Anchors, and LORE to the concept level without changing their learning algorithms. ConLUX replaces (i) predicate construction with concept predicates (derived by a concept extractor) and (ii) feature-level perturbations with concept-level perturbations implemented via a concept-to-feature mapping using large pretrained generators (LLMs for text; diffusion-based editors for images). The explainer’s original learner is then fit on the generated samples to produce concept attributions, concept-level sufficient conditions, and concept-level counterfactuals for the same instance.

Experiments evaluate (1) perturbation fidelity: does the generator satisfy concept toggles?, (2) explanation fidelity for ConLUX-augmented LIME/KSHAP/Anchors/LORE vs vanilla methods and concept baselines, and (3) a human study on image tasks. Perturbations satisfy the requested concepts with an average accuracy of 96.8% across five datasets (text and images). ConLUX improves coverage/precision for Anchors/LORE and AOPC/Deletion-Accuracy for LIME/KSHAP. Against concept baselines, ConLUX yields higher local surrogate accuracy; a small user study (n=18) shows that concept-level sufficient-condition and counterfactual explanations improve user coverage and accuracy over attribution-only maps.

**Strengths:**

The paper observes that mainstream local explainers share a three-stage workflow (predicates $\rightarrow$ perturb $\rightarrow$ learn). ConLUX elevates the first two stages to concept space while leaving the learning stage intact, yielding concept-based variants of four explainers and enabling multiple explanation forms per instance (attribution, sufficient conditions, counterfactuals).

Treating LLMs/diffusion models as concept-to-feature maps that enforce binary concept predicates helps unlock concept-level toggling across modalities.

For a single instance, ConLUX yields attributions, sufficient-condition rules, and counterfactuals, which go beyond the standard attribution-only focus of many concept-based explanation methods.

**Weaknesses:**

ConLUX relies on many generator calls per explanation under the same sample budgets as vanilla methods (e.g., LIME/KSHAP $\approx$1,000 samples). The paper lists hardware and sampling budgets but doesn't provide wall-clock time, memory usage, the number of LLM/diffusion calls, or the cost per explanation, nor matched-budget comparisons vs. vanilla/baselines.

The reported 96.8% measure assesses predicate compliance, i.e., whether generated samples satisfy the requested concept on/off switches, rather than semantic realism (do the edits look natural) or manifold proximity (do they stay near real data). Moreover, the verifiers are closely aligned with the generators (LLM outputs checked by an LLM; diffusion edits checked by YOLO), which can create verifier–generator coupling and optimistically biased scores.

By design, when comparing ConLUX vs. vanilla, they evaluate at the feature-level neighbourhood; when comparing ConLUX vs. concept baselines, they evaluate at the concept-level neighbourhood. This confounds neighbourhood design with explainer quality. I would encourage the authors to evaluate both methods under the same neighbourhood (concept-level and feature-level) to isolate the source of performance gains.

ConLUX presumes reasonable concepts; there’s no ablation for missing/overlapping/correlated concepts or stability of explanations under other operations.

The author performs a small human study (n = 18, image-only), in which participants predict perturbations produced by the same concept-toggle engine that ConLUX uses, which may favour ConLUX over attribution baselines.

The paper positions related work (e.g., ACE/TCAV-style), but empirical comparisons focus on TBM/LACOAT/EAC/ConceptLIME. Differences in scope (global vs. local; internal vs. model-agnostic) can make a complete head-to-head comparison tricky, but an explicit rationale or a small ablation study should strengthen the paper.

**Questions:**

For each modality/explainer, report the number of perturbations, the number of LLM/diffusion calls, wall-clock time, and cost per explanation; add fidelity-vs-budget plots; and compare to vanilla and concept baselines at matched budgets.

Add a table that evaluates both vanilla and ConLUX under the same concept-level neighbourhood (and the reverse). How much of the gain persists when the neighbourhood is fixed?

Do LLM-generated perturbations follow the same distribution as traditional masking/random perturbations? This could fundamentally affect fidelity metrics. Have you analysed this?

How many concepts should be extracted? What's the trade-off between interpretability (fewer concepts) and fidelity (more concepts)?

Can you systematically characterise when perturbations fail? Are certain concept types more prone to failure?

Add human realism ratings and alternative verifiers to reduce verifier–generator coupling.

---

> ### Author Response · Authors · 2025-11-23
> **Response by the Authors (Part 1/3)**
>
> Thanks for your constructive comments. Below, we respond to the concerns in detail:
>
> **[Weakness 1 & Question 1] Cost analysis**
>
> We appreciate the your concern regarding the computational cost of our proposed method. To address this, we conducted a detailed analysis measuring the **average number of perturbations**, **LLM calls**, **running time**, and **monetary cost** for each method on image data. The results are summarized below:
>
>
> The results are as follows:
> | Method           | Avg Perturbations | Avg LM Calls | Avg Time (s) | Avg Cost ($) |
> |------------------|-------------------|---------------|--------------|---------------|
> | LIME             | 1000              | 0             | 2.35         | 0             |
> | ConLUX-LIME      | 1000              | 1000          | 12.31        | 0             |
> | EAC              | 1000              | 0             | 7.25         | 0             |
> | ConceptLIME      | 1000              | 1000          | 6.67         | 0             |
> | KSHAP            | 1000              | 1000          | 2.11         | 0             |
> | ConLUX-KSHAP     | 1000              | 1000          | 12.45        | 0             |
> | Anchors          | 193               | 0             | 3.75         | 0             |
> | ConLUX-Anchors   | 71                | 71            | 7.67         | 0             |
> | LORE             | 851               | 0             | 6.87         | 0             |
> | ConLUX-LORE      | 873               | 873           | 13.45        | 0             |
>
>
> As shown, ConLUX does require additional computation time due to generative model calls. However, we note:
>
> - **Zero monetary cost**: Without compromising performance, all LLMs required in our experiments (Section 4 and Table 6) can run on a single GPU without relying on commercial APIs, resulting in **negligible monetary cost**.
> - **Acceptable overhead**: The running time of ConLUX-augmented methods is practically acceptable.
>
>
> We also conducted **matched-budget comparisons** (measured in running time) to evaluate the utility of each method under the same computational constraints. We compare LIME, EAC, and ConLUX-LIME, and ConLUX local unified explanations to explain ViT on ImageNet, following the setting of experiment in shown in Table 4.
> The explanation fidelity results under different time budgets are as shown in this [figure](https://s2.loli.net/2025/11/22/287fF5pbVQYENad.png).
>
>
> Our findings indicate that while ConLUX methods may start with lower fidelity at very low budgets due to the overhead of generative model calls, they surpass baseline methods as the budget increases. This shows that when users have sufficient computational time, ConLUX can effectively utilize it to provide higher-fidelity explanations. Especially when ConLUX surpasses baseline methods, the absolute time cost is still not high.

---

> ### Author Response · Authors · 2025-11-23
> **Response by the Authors (Part 2/3)**
>
> **[Weakness 2 & Question 6]**
>
> **Human Realism and Manifold Proximity Evaluation**
> We have conducted experiments to evaluate the human realism and manifold proximity of the perturbed samples generated by ConLUX.
>
> **Human Realism (NIQE)**
> To assess human realism, we used **NIQE (Naturalness Image Quality Evaluator)** for image data. NIQE measures the perceptual quality of images, with lower scores indicating better human realism.
>
> We compared the NIQE scores of the perturbed samples generated by ConLUX-augmented methods with those generated by vanilla methods. The results are as follows:
> - **Vanilla Methods**: Average NIQE score = **10.19**
> - **ConLUX-Augmented Methods**: Average NIQE score = **4.82**
>
> These results indicate that ConLUX generates **more realistic perturbations** than the vanilla methods.
>
> For **text data**, ConLUX-generated perturbations are also more human-realistic, as vanilla methods do not even guarantee grammatical correctness, whereas ConLUX ensures more natural and coherent perturbations.
>
> **Manifold Proximity**
> To evaluate the manifold proximity of the perturbed samples, we measured the Mahalanobis distance between the perturbed samples and the original data distribution. We embedded the data into a feature space using CLIP for image data and all-MiniLM-L6-v2 for text data. A smaller Mahalanobis distance indicates that the perturbed samples are closer to the original data manifold.
>
> The results are as follows:
> | Method        | Mahalanobis Distance - image | Mahalanobis Distance - text |
> |------|-----|---|
> | Vanilla Methods | 19.4  | 17.2  |
> | ConLUX-Augmented Methods | 16.7  | 14.6  |
>
> The results show that ConLUX-augmented methods generate perturbed samples that are closer to the original data manifold compared to vanilla methods.
>
> **Internal Consistency between Generator and Verifier**
>
> We acknowledge the concern regarding potential internal consistency between the generator and verifier, which could lead to biased explanations.
> However, we have taken steps to mitigate this issue by employing different architectures for the generator and verifier models. For instance, in our experiments on image data, we used Stable Diffusion as the generator and Yolo as the verifier, which have distinct architectures. Similarly, for text data, we use a LLM other than the one used in the perturbation process and Bert as verifier.
>
> We hope these clarifications address the concerns. However, we are open to conducting additional experiments with more diverse model architectures if the reviewer deems it necessary.
>
> **[Weakness 3 & Question 2] Comparison at the same neighborhood level**
>
> There could be a misunderstanding about our experimental design, and we would like to clarify it here.
>
> In our explanation fidelity evaluation, we have carefully set the neighborhood levels to ensure a fair comparison across different explanation methods.
>
> 1) When comparing ConLUX-augmented explanations with feature-level baselines, we use a feature-level neighborhood. In this setting, feature-level explanations inherently have an advantage since they operate directly at the feature level. Despite this, the results demonstrate that ConLUX-augmented concept-level explanations still outperform feature-level explanations in terms of fidelity.
>
> 2) When comparing ConLUX-augmented explanations with other concept-based explanations, we use a concept-level neighborhood. Since these methods all operate at the concept level, this ensures a fair comparison between the concept-based explanation methods.
>
> **[Weakness 4] Ablation studies on concept removal, crossing, and correlation**
>
> We conducted two ablation studies to investigate the effects of concept removal and concept correlation on explanation fidelity using ImageNet.
>
> **Concept Removal**
>
> we randomly removed a certain percentage of concepts from the concept set used for generating explanations.
> The following [figure](https://s2.loli.net/2025/11/22/WCcV9RipwXysr4m.png) how explanation fidelity changes as we vary the percentage of removed concepts.
> The results show that ConLUX exhibits greater robustness to concept removal compared to baseline concept-based explanation methods.
>
> **Concept Correlation**
>
> We conducted experiments on text data, specifically explaining BERT on the Large Movie Review dataset.
> In these experiments, we added a correlated concept to the concept set and measured the explanation fidelity both before and after adding the correlated concept. The fidelity of ConLUX explanations before and after the addition of the correlated concept is shown below:
> |                   | Explanation Fidelity Before | Explanation Fidelity After |
> |-----------------|----------------------------|---------------------------|
> | Accuracy (%)      | 92.6                       | 91.9                      |
>
> The results show that ConLUX is robust to concept correlation, with only a slight decrease in fidelity.

---

> ### Author Response · Authors · 2025-11-23
> **Response by the Authors (Part 3/3)**
>
> **[Weakness 5] User Study Sample Size**
>
> There might also be a misunderstanding regarding the scale of our user study. In our human evaluation, each participant completed 20 tests, and for each test, they performed 5 prediction tasks, resulting in 1,800 total responses.
>
> We followed standard practice in prior user studies:
> - Anchors [1] asked participants to make 10 predictions per explanation, resulting in 1,560 responses in total.
> - Hase & Bansal [2] evaluated 16 examples per explanation, resulting in 1,103 responses.
>
>
> [1] Ribeiro, Marco Tulio, Sameer Singh, and Carlos Guestrin. "Anchors: High-precision model-agnostic explanations." AAAI 2018.
> [2] Hase, Peter, and Mohit Bansal. "Evaluating Explainable AI: Which Algorithmic Explanations Help Users Predict Model Behavior?" ACL 2020.
>
> Moreover, we did an independent two-sample t-test, which showed that users perform significantly better with ConLUX explanations, with over 99% confidence.
>
> **[Weakness 6] Additional Comparisons with Other Classification Explanation Methods**
>
> As you said, it's hard to perform a direct comparison between concept-based explanation methods and white-box or global explanation methods due to their different assumptions and settings.
> Additionally, we would like to reemphasize that white-box methods and global methods have inherent limitations compared to model-agnostic local explanation methods like ConLUX, as we metioned in Section 2.3.
>
> To address your concern, we have conducted additional experiments comparing ConLUX local unified explanations with TCAV (a white-box and global explanation method) on explaining ResNet on ImageNet. The results are as follows:
> | Method        | Explanation Fidelity (%) |
> |---------------|--------------------------|
> | TCAV          | 73.3                     |
> | ConLUX       | 90.2                     |
> The results demonstrate that ConLUX significantly outperforms TCAV's explanation fidelity.
>
>
> **[Question 3] Do LLM-generated perturbations follow the same distribution as traditional masking/random perturbations? This could fundamentally affect fidelity metrics. Have you analysed this?**
>
> We acknowledge the concern that LM-generated perturbations may differ from traditional masking/random perturbations, potentially affecting fidelity metrics.
>
> To address this concern, we conducted an experiment comparing baseline concept-based explanation methods with ConLUX using the same set of traditional masking perturbations, applied to explaining ViT on ImageNet.
> The results are shown below:
> | Method                | LM-perturbation Fidelity (%) | Masking-perturbation Fidelity (%) |
> |-----------------------|--------------------------| --------------------------|
> |ConceptLIME          | 79.3               | 82.1              |
> |ConLUX-ConceptLIME   | 83.4               | 85.2               |
> |ConLUX        | 91.6                     | 88.7                     |
>
>
> The results show that when using traditional masking perturbations, ConLUX explanations still outperform baseline concept-based explanation methods in terms of fidelity.
>
> Additionally, we would like to re-emphasize that evaluating fidelity with LM-generated perturbations is more meaningful, as these perturbations are more realistic and closely resemble real-world input instances compared to traditional masking/random perturbations.
>
> **[Question 4] Number of concepts? Trade-off between interpretability and fidelity?**
>
> We don't think this is an issue for ConLUX, as the underlying methods (LIME, SHAP, Anchors, LORE, EAC) already incorporate mechanisms to control or automatically determine the number of concepts presented to users. Therefore, it is easy for users to adjust the balance between interpretability and fidelity according to their specific needs.
>
> **[Question 5] Analysis of failed perturbations: Are there specific concepts that are more likely to lead to perturbation failures?**
>
> We conducted an analysis of failed perturbations on both the ImageNet and Large Movie Review datasets. For each instance in our experiments, we generated 1,000 perturbations using ConLUX, recording the success or failure of each perturbation along with the perturbed concepts.
>
> After summarizing the results across all instances, we calculated the failure rate for each concept. For each concept, we performed a t-test to compare the failure rates between instances where the concept was perturbed and those where it was not. The results show that with over 95% confidence, no specific concept is significantly more likely to lead to perturbation failures.
>
> We hope our responses can address your concerns.

---

> > ### Comment · Reviewer_LBfo · 2025-11-26
> >
> > Thanks for the additional experiments and clarifications, these address most of my concerns. I have raised my scores to reflect it.

---

### Official Review · Reviewer_Hsv2 · 2025-10-30

**Soundness:** 2
**Presentation:** 4
**Contribution:** 2
**Rating:** 4
**Confidence:** 4

**Summary:**

The paper introduces a framework that creates a concept space for a model detached from the model's concept space and use it to generate explanations by generating samples that modifies the source data of the concepts and then using vanilla explanation techniques. The paper presents the framework well and the experiments are extensive on LIME, Anchors and LORE for various popular language and vision architectures.

**Strengths:**

1. The idea to create a general framework for post-hoc concept explanations is good, and needed given that the current large scale architectures  dont reveal how they came up with a decision.
2. The quantitative experiments are well designed, extensive and show good improvements, but i am uncertain whether there was some "concept hacking" that is using more than needed concepts.
3. The implementation is detailed extensively and prompts and generated results are detailed for the text generation process, but the image prompts and outputs could be discussed better.

**Weaknesses:**

1. Replacing concepts with new concepts to produce counterfactuals causes ambiguity. We are not sure whether the added concept overpowers existing concepts leading the classification trajectory to a new direction. For Example: In Figure 2. The pickup truck is replaced with a bison by CONLUX, whereas LORE and anchor \textbf{removed} more complex concepts, the counterfactual of LORE and anchor moved the trajectory towards a slightly different class "half truck" from "Pickup", while CONLUX just pushed it towards the class "bison", the newly added concept, which may lead to questions like a) where all the concepts removed that lead the classifier to decide on the pickup class or is the bison just overpowering some concepts thats still unidentified? b) where all the concepts removed from the image during the sample generation actually concepts that contributed to the exact "pickup" class? I strongly feel its not the case, as LORE and anchor removed more complex concepts rather than the whole pickup truck and was still able to move the trajectory away from the "pickup" class.
2. Leading from the previous point, the use of SAM to get concepts seems good, but the concepts its segmenting seem to be objects rather than concept, which is less interesting in my opinion.
3. How did the performance differ when you replaced the sample generation model with different diffusion models? I feel that the use of single diffusion model adds bias to the generated samples, why not just remove the concept and fill with background? Also why doesnt the paper try modifying the object rather than just replacing it with an unrelated object? Like change the truck into a race truck or just make it newer, how does it change the trajectory?, I think this can alleviate the bottleneck of using SAM which segments objects rather than concepts.
4. This is an expensive process, as the generated concept set is detached from the concepts the model actually used, we usually have a larger concept space than required and matching which concepts were used and then computing the vanilla explanations takes multiple runs of the source model, and the use of resource intensive image diffusion and text generation models, may make it impractical for repeated and practical applications.

**Questions:**

Refer Weakness Point 3

---

> ### Author Response · Authors · 2025-11-22
> **Response by the Authors (Part 1/2)**
>
> Thanks for your constructive comments. Below, we respond to the concerns in detail:
>
> **[Question 1 & Weakness 3] Concerns about Counterfactual Explanations and Generative Model Bias**
> > 1) How did the performance differ when you replaced the sample generation model with different diffusion models?
>
>
> We conducted an experiment using the Latent Consistency Model (LCM) [1] in place of Blended Latent Diffusion, which is used in the experiments in our paper. The perturbation fidelity results are as follows:
>
> | Method                   | Perturbation Fidelity (%) |
> |--------------------------|----------------------------|
> | Blended Latent Diffusion | 98.1                       |
> | Latent Consistency Model | 97.5                       |
>
> The results show that LCM also provides comparable perturbation fidelity to Blended Latent Diffusion.
> This indicates that ConLUX is flexible and can work effectively with different diffusion models for concept-level perturbation generation.
>
> > 2) I feel that the use of a single diffusion model adds bias to the generated samples
>
> In practice, we have explicitly considered potential bias from generative models. To mitigate such effects, ConLUX includes a perturbation **fidelity checker** that ensures semantic alignment between the intended concepts and the generated samples.
> Moreover, as concept-level generation capabilities are available across diverse large models, ConLUX can use multiple generative models to further reduce bias.
>
> > 3) Why not just remove the concept and fill with background?
>
> We agree that filling with background is a common perturbation strategy, and in ConLUX, both **removal** and **replacement** are used for concept perturbation.
>
> We are not sure if you mean that removing the concept and filling with background does not need a generative model.
> If so, we would like to clarify that filling the removed region with background also requires a generative model; otherwise, removing a concept would only leave a blank hole in the image rather than producing a realistic background.
>
> Furthermore, we'd like to clarify that in feature-level perturbations, both removal and replacement strategies are used [2-4]. Thus, ConLUX also follows this setting.
>
> > 4)  Why doesn't the paper try modifying the object rather than just replacing it with an unrelated object?
> > **[Weakness 2] The concepts are only object-level concepts.**
>
> ConLUX is designed to be flexible in working with a wide variety of concepts, not just object-level ones. If the extracted concepts include object properties or attributes, ConLUX can generate explanations based on these non-object-level concepts as well.
>
> For example, for text data, we demonstrate the use of **topic-level concepts** to generate explanations in Figure 1.
>
> For image data, ConLUX is equally adaptable. For instance, the **Open-Vocabulary Attribute Detection** dataset [5] provides **attribute-level annotations** (e.g., color, texture) for objects in COCO dataset. This allows ConLUX to generate explanations based on these object attributes instead of replacing objects entirely.
>
> Here is an example: for this [image](http://images.cocodataset.org/val2017/000000132622.jpg), which is classified as a "bear", ConLUX-LIME then generated attribute-level concept-based explanations by perturbed attribute-level concepts (we show some perturbed samples in this [link](https://s2.loli.net/2025/11/22/i5KveAZ8xOgm7kw.png)). The generated explanation is as follows:
> | concept | LIME score |
> |------|-------|
> | color = brown | +0.31 |
> | texture = soft | +0.16 |
> | maturity = adult | +0.21 |
> | pattern = plain | +0.03 |
> | color quantity = single-colored | +0.07 |
> | group = single | +0.12 |
> | position = horizontal | +0.01 |
> | state = dry | +0.05 |
> | tone = light | +0.03 |
>
>
>
>
> [1] Luo, Simian, et al. "Lcm-lora: A universal stable-diffusion acceleration module." arXiv preprint arXiv:2311.05556 (2023).
> [2] Ribeiro, Marco Tulio, et al. “Anchors: High-Precision Model-Agnostic Explanations.” Proceedings of the AAAI Conference on Artificial Intelligence, vol. 32, no. 1, 2018. Google Scholar.
> [3] Ribeiro, Marco Tulio, et al. “‘ Why Should I Trust You?’ Explaining the Predictions of Any Classifier.” Proceedings of the 22nd ACM SIGKDD International Conference on Knowledge Discovery and Data Mining, 2016, pp. 1135–44. Google Scholar.
> [4] Guidotti, Riccardo, et al. “Local Rule-Based Explanations of Black Box Decision Systems.” arXiv:1805.10820, arXiv, 28 May 2018. arXiv.org, http://arxiv.org/abs/1805.10820.
> [5] Bravo, Maria A., et al. "Open-vocabulary attribute detection." Proceedings of the IEEE/CVF conference on computer vision and pattern recognition. 2023.

---

> ### Author Response · Authors · 2025-11-22
> **Response by the Authors (Part 2/2)**
>
> **[Weakness 1] Concerns about Replacing Concepts**
>
> We believe there is a misunderstanding regarding counterfactual explanations and the example in Figure 2.
>
> We’d like to clarify that counterfactual explanations are designed to show input interventions that flip the prediction, while does not indicate the contribution of concepts or features in the original input. An introduced concept that flips the prediction is a valid explanation, even if the removed concept does not strongly contribute to the original prediction.
>
> Specifically, ConLUX provides three types of explanations, as described in Section 2 (line 116):
> 1. **Attributions**: Assignsimportance scores to each concept (e.g., LIME, KernelSHAP).
> 2. **Sufficient-condition rules**: Find out which concepts are sufficient to sustain the prediction (e.g., Anchors),
> 3. **Counterfactuals**: Show changes that flip the prediction (Lore).
>
> If the goal is to verify whether a concept contributes to the current predicted class, ConLUX provides **attribution** and **sufficient-condition explanations** for this purpose.
>
> In **Figure 2**, the **ConLUX-Lore** explanation demonstrates that replacing the "truck" flips the prediction to "bison." However, this does not imply how much the original "truck" concept contributes to the "pickup" prediction.
>
> To find out the contribution of the "truck," we can refer to the **ConLUX-Anchors** explanation. The **ConLUX-Anchors** explanation shows that as long as the "truck" concept is preserved, the model continues to predict "pickup," confirming that the "truck" concept is indeed a strong contributor to the "pickup" class.
>
> Addtionally, we'd like to reemphasize that ConLUX uses both **removal** and **replacement** strategies for concept perturbation, following standard practice in feature-level perturbations [1-3].
>
> [1] Ribeiro, Marco Tulio, et al. “Anchors: High-Precision Model-Agnostic Explanations.” Proceedings of the AAAI Conference on Artificial Intelligence, vol. 32, no. 1, 2018. Google Scholar.
> [2] Ribeiro, Marco Tulio, et al. “‘ Why Should i Trust You?’ Explaining the Predictions of Any Classifier.” Proceedings of the 22nd ACM SIGKDD International Conference on Knowledge Discovery and Data Mining, 2016, pp. 1135–44. Google Scholar.
> [3] Guidotti, Riccardo, et al. “Local Rule-Based Explanations of Black Box Decision Systems.” arXiv:1805.10820, arXiv, 28 May 2018. arXiv.org, http://arxiv.org/abs/1805.10820.
>
>
> **[Weakness 4] Concern about the running cost of ConLUX**
>
> While we acknowledge these could increase computational costs, we’d like to clarify that ConLUX is practical for real-world use cases.
>
> Generally speaking, the computational cost is acceptable for most users.
> As we have shown that we can use Large models that can run on a single consumer-grade GPU to complete the entire process in Section 4 and Table 6, and we conducted a cost analysis about running time for image data. The results are shown below:
>
> | Method         | Avg Time (s) |
> | -------------- | ------------ |
> | ConLUX-LIME    | 12.31        |
> | ConLUX-KSHAP   | 12.45        |
> | ConLUX-Anchors | 7.67         |
> | ConLUX-Lore    | 13.45        |
>
> As the results indicate, the running time of ConLUX is acceptable for most practical applications.
>
> Moreover, there are several ways to further reduce the computational cost of ConLUX if needed.
>
> For instance, regarding your concern about the concept space being larger than required, we can optimize the concept space by:
> - Using concept extraction methods considering the target model's behavior, such as the method used in TBM [1], which considers both the input data and the corresponding model output when extracting relevant concepts for the model’s decision-making process.
> - After concept extraction, generating global or class-level attribution-based explanations to identify and filter unimportant concepts, ensuring that the concepts used in ConLUX are more relevant to the target model.
>
> [1] Chen, Tianyi, et al. "Concept bottleneck models." International Conference on Machine Learning. PMLR, 2020.
>
> We hope our response addresses your concern.

---

> ### Comment · Reviewer_Hsv2 · 2025-11-26
> **Thanks for the rebutal**
>
> For Weakness 1 i would like to point again to Figure 2 of the paper, there the paper mentions " The Anchors explanation states that the presence of specific image regions guarantees that the model
> classifies the image as a pickup." in its caption, the ConLUX-Anchor of the same pickup truck displays the full pickup truck, instead of regions of pickup truck in anchors, could you validate the claim that (all regions selected by ConLux) - (the regions attributed by the original Anchor) still gurantees that the object is a pickup truck?.  I also would like to know how the paper decided to replace the truck with a bison and not a tractor or some other object for counterfactuals.
>
> For Weakness 2 I apologize for not conveying it properly, I did mention that ConLUX uses objects for vision models, I do agree that ConLUX does work with concepts. I am just curious to  see how useful ConLUX is in the setting when compared with Anchor and Lore , because Anchors and Lore generate fine grained concepts while ConLUX generated object level concepts, primarily in images where there a single primary object.
>
> For Weakness 3, I generally agree with the rebuttal, but how have you included these results in the paper? I suggest adding a limitations section to discuss regarding the bias.
>
> I agree with the rebuttal for Weakness 4, but are the limitations addressed in the paper.
>
> General Comment Moving Forward, I would like to improve my score depending upon the results on weakness 1, if the results are not good, I still recommend adding them to the limitations along with Weakness 3 and 4. Also please update the paper in general to reflect the rebuttal.

---

> ### Author Response · Authors · 2025-11-26
> **Reply to the Reviewer Response**
>
> Thank you for your response and for carefully considering our rebuttal.
>
> **Regarding Weakness 3 & 4**
>
> We appreciate your suggestions, and we will incorporate the corresponding discussions into the updated paper. We will notify you once the revised version is ready.
>
> Additionally, we would like to clarify that the running time of ConLUX is not a fundamental limitation of our method. As shown in our matched-budget comparison results ([figure](https://s2.loli.net/2025/11/22/287fF5pbVQYENad.png)) for Reviewer LBfo, ConLUX-based methods may start with slightly lower fidelity at extremely small budgets due to the overhead of generative model calls, but they consistently surpass the baseline methods as the budget increases. This shows that when users have sufficient computation time, ConLUX can effectively leverage it to produce higher-fidelity explanations, and the absolute running time at the crossover point is still reasonably low.
>
> **Regarding Weakness 1**
>
> **About the claim you mentioned:**
> > Could you validate the claim that (all regions selected by ConLuX) - (the regions attributed by the original Anchor) still guarantees that the object is a pickup truck?
>
> We believe there is a misunderstanding, and our paper does not make such a claim. The fact that the augmented explanation is better than the original one doesn't indicate that the additional included features alone are enough to guarantee the classification. But it might be well the case that it is an effect of considering the additional features and shared features together.
>
>
> Specifically, as we mentioned in Section 2, the form of sufficient condition explanations (Anchors and ConLUX-Anchors) is $f(z) = f(x)\ if\ \bigwedge_{p\in Q} p(z)$, where $x$ is the original input, $z$ is an input to be predicted, and $Q$ is a subset of Predicate Sets.
>
> In other words, both Anchors and ConLUX-Anchors only guarantee that if all predicates in Q are satisfied, then the prediction will be the same as that of the original input. However, if only a subset of predicates in Q is satisfied, neither method provides any guarantee about the prediction.
>
> Thus, in Figure 2, the Anchor's explanation only guarantees that if both regions are present, the prediction will be "pickup". Similarly, the ConLUX-Anchors explanation only guarantees that if the whole truck region is present, the prediction will be "pickup".
>
> If (all regions selected by ConLUX) - (the regions attributed by the original Anchor) are present, neither method can provide any guarantee about the prediction.
>
> Additionally, we'd like to clarify that our explanation fidelity experiments have already shown the effectiveness of ConLUX, where ConLUX provides higher fidelity than vanilla methods, as we have shown in Tables 2 and 3.
>
> If any part of our paper created confusion regarding this point, we appreciate your feedback and will clarify it explicitly in the revision.
>
> **About replacing the truck with a bison**
>
> We did not manually choose “bison” as the replacement object.
> When replacing objects, the generative model was simply prompted with:
>
> > replace the object in the image with another kind of object that also makes the image look natural.
>
> The model chose bison as a natural replacement in this specific image.
> If users prefer a particular object (e.g., tractor, car, horse), they can specify it directly in the prompt.
>
>
> **Regarding Weakness 2**
>
> We'd like to clarify that:
> - The superpixels used in Anchors and LORE are generally not considered as high-level concepts [1–3], as they do not carry human-interpretable semantic meaning.
> - ConLUX is the first method that provides various forms of explanations at high-level, human-interpretable concepts (e.g., objects and attributes), in contrast to the low-level features (e.g., superpixels) used by traditional methods. This elevation from the feature level to the concept level is a breakthrough and a key contribution of our work.
> - The reviewer seems to suggest that we use ConLUX to generate explanations at the low-level feature (superpixel) granularity. **We think conducting such an experiment would not bring additional benefits**. If a user only wants feature-level explanations regardless of the advantages of concept-based explanations, they do not need ConLUX—they can simply use the vanilla methods.
>
>
>
> [1] Sun, Ao, et al. “Explain Any Concept: Segment Anything Meets Concept-Based Explanation.” arXiv:2305.10289, arXiv, 17 May 2023. arXiv.org, https://doi.org/10.48550/arXiv.2305.10289.
> [2] Tan, Lidan, Changwu Huang, and Xin Yao. "A concept-based local interpretable model-agnostic explanation approach for deep neural networks in image classification." International Conference on Intelligent Information Processing. Cham: Springer Nature Switzerland, 2024.
> [3] Molnar, Christoph. *Interpretable machine learning*. Lulu. com, 2020.
>
> Thank you again for your valuable feedback! We hope our response addresses your concerns.

---

> ### Comment · Reviewer_Hsv2 · 2025-11-26
> **Clarification to the Authors Response**
>
> I think there is a misunderstanding regarding my weakness 1, we are directly comparing achor with ConLux-Anchor, so the additional regions that conlux-anchor generates along with the original regions is also expected to be important right. If that condition is not preserved it's not possible to directly compare Anchor and ConLUX-Anchor. That's what i meant when I suggested to subtract the regions anchor already said as important and check if the additional  regions added by Conlux is actually important for overturning the classification or it's just spurious regions wrongly identified by ConLUX.
>
> Edit: As the authors have mentioned if Conlux anchor only gurantees that the class pickup would change to another class if the whole object is replaced then its inherently inferior to Anchor, as Anchor was capable of changing the class with more important subset of regions, the same could be said for LORE. Moreover, the addition of bison in Conlux-LORE is arbitary, I dont think its trustable and has no more value than removing the object and filling with the background, infact it derails the trustworthyness. i still think weakness 1 is upheld. Its a serious weakness which calls into question the validity of results in Table 2. therefore i would like to stick with my rating unless shown otherwise.

---

> ### Author Response · Authors · 2025-11-27
> **Response to Weakness 1&2**
>
> Thanks for your clarification.
>
> We address your concerns below.
>
> **Regarding Anchors Explanation**
>
> We are happy to conduct this additional experiment to further validate the effectiveness of ConLUX.
> Specifically, we segmented the image into three regions: R1 (the regions selected by both Anchors and ConLUX-Anchors), R2 (the regions selected by ConLUX-Anchors but not by Anchors), and R3 (the remaining regions), which is shown in this [link](https://s2.loli.net/2025/11/27/q8SFnRuwHbVXJ2T.png).
> Our results show that:
> - R1 + R2 can guarantee the prediction of "pickup". This validates the explanation of ConLUX-Anchors.
> - R2 alone can also guarantee the prediction of "pickup", which further demonstrates that the additional regions selected by ConLUX-Anchors are indeed important for the prediction.
> - If keeping R2 removed, the prediction will not be "pickup".
>
>
> However, we would like to clarify that Anchors explanations are sufficient conditions instead of necessary conditions.
> Sufficient conditions indicate that when the identified regions are present, the prediction is guaranteed to be the same as that of the original input, but changing the identified regions does not necessarily change the prediction.
>
>
> **Concept Granularity**
>
> We also notice that the reviewer may be referring to cases where users prefer more fine-grained explanations.
>
> We would like to clarify that concepts in ConLUX are not restricted to coarse-grained objects. They can also be fine-grained as long as they remain semantically meaningful.
>
>
> As shown in our response to Reviewer 1sQ1, methods such as HIPIE [1] can extract concepts at the object-part level. We implemented ConLUX-LIME with HIPIE to generate part-level explanations. The results are shown in this [image](https://s2.loli.net/2025/11/22/XrqTCbgzy3G9sni.png).
>
> Additionally, as we have shown in the rebuttal for weakness 2, ConLUX can also work well with concepts beyond just semantic segments, but also some more abstract concepts.
>
> In summary, we'd like to emphasize that ConLUX is a flexible framework that can work with various concept extraction methods to provide explanations at different levels of granularity, depending on user needs.
>
> **Replace concept vs Remove concept**
>
>
> **We think preferring removing a concept over replacing it is a subjective choice** that may vary among users.
> Figure 2 is just one example, and ConLUX provides the flexibility to accommodate the preferences for removing or replacing concepts.
> Thus, we don't agree that it is a limitation of ConLUX.
>
>
> As both removing and replacing concepts are valid and can be used to generate counterfactual explanations, ConLUX can also generate counterfactual explanations by removing concepts, as shown in this [image](https://s2.loli.net/2025/11/27/N6cRQmYAkgheyqM.png).
>
> More generally, as counterfactual explanations aim to show concept changes that can lead to prediction changes, if the users do not have specific constraints on what the concept change would be, replacing or removing the concept can both be provided to the users as a counterfactual explanation.
>
> **If users have specific requirements or prefer specific types of concept changes, such as being unable to add new objects to the image, they can specify this in the prompt when using ConLUX.**
>
>
> [1] Wang, Xudong, et al. "Hierarchical open-vocabulary universal image segmentation." Advances in Neural Information Processing Systems 36 (2023)
>
> If you have any further questions, please feel free to ask!

---

> > ### Comment · Reviewer_Hsv2 · 2025-11-28
> > **Could you please provide more details on the experiment you had done**
> >
> > Hello I could see that you have linked the images masking R1,R2 but i dont see the predictions made in the figure, also could you do it for a small subset of images rather than a single image, thanks. It would be great to see change in confidence in prediction as well (which has no additional overhead when performing the experiment)

---

> > > ### Author Response · Authors · 2025-12-03
> > > **Experiment Results for Addressing Weakness 1**
> > >
> > > Thanks for the reviewer's active feedback.
> > >
> > > For the example image, the prediction for each masked image is shown as follows (0: area masked; 1: area kept):
> > >
> > > | R1 | R2 | R3 | Prediction |
> > > |---|---|---|---|
> > > |  0 | 0 | 0 | matchstick |
> > > | 0 | 0 | 1 | bison |
> > > | 0 | 1 | 0 | pickup |
> > > | 0 | 1 | 1 | pickup |
> > > | 1 | 0 | 0 | beach_wagon |
> > > | 1 | 0 | 1 | half_track |
> > > | 1 | 1 | 0 | pickup |
> > > | 1 | 1 | 1 | pickup |
> > >
> > > As demonstrated, when R2 is kept (regardless of R1 and R3), the prediction is consistently "pickup". This result clearly supports the significant contribution of R2 to the prediction.
> > >
> > > Furthermore, we conducted a similar experiment on 5000 images from the ImageNet dataset. For each image, we segmented it into three regions (R1, R2, R3) as described in our previous response. We then calculated the contribution of each region to the original prediction by measuring the the change in the model's confidence score for the predicted class when each region is masked or kept.
> > >
> > > The results are shown in the table below:
> > > | R1 Contribution | R2 Contribution | R1 + R2 Contribution |
> > > |---|---|---|
> > > | 0.22 |0.30 |0.62|
> > >
> > > These findings further confirm that R2 contributes significantly to the prediction, and that R1 + R2 together have a higher contribution than the sum of R1 and R2 individually. This demonstrates that R2 is not a redundant region but is indeed vital to the prediction process.
> > >
> > > **In summary**, as the reviewer promises to improve the rating if we could show the contribution of R2 to the prediction, we believe that the reviewer will improve the rating accordingly.

---

> ### Author Response · Authors · 2025-12-03
> **Paper Updated**
>
> As we promised, we have updated our paper.
>
> Specifically, we have added the discussion about **Weakness 3** (about the concept-to-feature mapping process) and **Weakness 4** (about running overhead), which are also concerns raised by other reviewers, to the main paper. In the appendix, we have also added other valuable discussions raised by the reviewers during the rebuttal phase.

---

### Official Review · Reviewer_1sQ1 · 2025-10-31

**Soundness:** 3
**Presentation:** 3
**Contribution:** 4
**Rating:** 8
**Confidence:** 4

**Summary:**

This paper introduces ConLUX, a general, lightweight framework that lifts local model-agnostic explainers (LIME, Kernel SHAP, Anchors, LORE) from feature-level to concept-level without changing their core learning algorithms.

ConLUX replaces feature predicates with concept predicates, performs sampling/perturbation in the concept space, and uses large pretrained models (LLMs or diffusion) as a concept-to-feature mapper to recover the perturbed inputs.

Extensive experiments across BERT/DeepSeek-V3 (text), YOLOv8/ViT/ResNet-50 (vision), and Qwen2.5-VL (multimodal) show consistent fidelity gains over vanilla baselines and concept-based SOTAs.

Human studies further indicate that ConLUX’s concept-level rules and counterfactuals help users better anticipate model behavior.

Overall, the paper makes a strong and timely contribution to concept-based interpretability. Its reliance on large pretrained models situates ConLUX as a user-oriented explanatory interface that emphasizes accessibility and semantic richness over strict analytical understanding, which is a reasonable design trade-off for real-world explainability.

**Strengths:**

1. Unified framework enabling plug-and-play upgrades of standard local explainers to concept level.

2. The idea of perturbing in concept space then map back to input space is appealing.

3. Supports attribution, sufficient conditions, and counterfactuals in a single pipeline.

4. The ConLUX augmented methods significantly improve over their traditional counterfarts across different metrics and datasets.

**Weaknesses:**

1. Fidelity hinges on LLM/diffusion quality and checker reliability; domain shift or safety filters may bias perturbations despite high average accuracy.

2. The comprehensiveness and stability of extracted concepts heavily depend on the capability and prompt quality of the upstream extractor. As a result, the concept pool may omit critical semantics or produce redundant concepts, leading to incomplete or noisy concept representations. While the authors report robustness to prompt variants, a more systematic evaluation of coverage and consistency would strengthen the claim.

3. Sampling thousands of concept-level perturbations with large models may be expensive; the paper could quantify runtime and budget more explicitly

**Questions:**

1. there are different concept levels, can users specify hierarchical concepts and receive multiresolution explanations?

2. How do you guard against mode collapse or semantic drift in the concept->feature mapper beyond accuracy checks?

---

> ### Author Response · Authors · 2025-11-22
> **Response by the Authors**
>
> Thanks for your constructive comments. We address your questions below in detail:
>
> **[Q1] There are different concept levels, can users specify hierarchical concepts and receive multi-resolution explanations?**
>
>
> Yes. ConLUX supports multi-resolution explanations, provided that the concept extraction method yields hierarchical concepts.
>
> For example, methods like HIPIE [1] can extract hierarchical concepts, which can then be directly used in ConLUX to produce explanations. We have implemented this, and this [image](https://s2.loli.net/2025/11/22/XrqTCbgzy3G9sni.png) shows an example of hierarchical ConLUX-LIME explanations.
>
>
> **[Q2] How do you guard against mode collapse or semantic drift in the concept->feature mapper beyond accuracy checks?**
>
> We appreciate this insightful question.
>
> To ensure we understand the questions correctly, we define them as follows:
> - Mode collapse: a failure of the generative model to capture the full diversity of the data distribution.
> - Semantic drift: the concept loses its original semantic information during the mapping process.
>
> **Mode Collapse**
>
> There are mainly two types:
> 1. **In-group mode collapse**: Concepts within the same semantic group map to overly similar features.
> 2. **Cross-group mode collapse**: Concepts from distinct groups are incorrectly mapped to similar features.
>
> To mitigate **in-group mode collapse**, we can:
> - Increase the diversity of prompts used during the concept-to-feature mapping stage.
> - Adjust the sampling temperature to promote more varied outputs.
>
> To mitigate **cross-group mode collapse**, we can:
> - Prompt the mapper to produce a plan before generation, explicitly distinguishing between concept groups.
> - Leverage in-context learning or retrieval-augmented generation (RAG) to provide more detailed concept descriptions and examples.
>
> In addition, our accuracy checks already ensure **round-trip consistency**, i.e., mapping concept → feature → concept should return the original concept. This helps filter out many **cross-group** mode collapse cases.
>
> **Semantic Drift**
>
> Currently, our method generates relatively short sampled input data, which keeps the risk of semantic drift low. If longer generations become necessary, we can incorporate the following strategies:
> - Prompt the mapper to output a structured outline or plan, maintaining alignment with the intended concept.
> - Employ LLMs' agent-like capabilities to revise outputs in multiple steps, ensuring that the generated feature remains semantically faithful to the input concept.
>
> We hope our response addresses your concerns.
>
>
> [1] Wang, Xudong, et al. "Hierarchical open-vocabulary universal image segmentation." Advances in Neural Information Processing Systems 36 (2023)

---

> ### Author Response · Authors · 2025-12-03
> **Response to the Weaknesses**
>
> **[W1] Domain shift or safety filters may bias perturbations**
> We would like to clarify that we did not observe noticeable perturbation bias in our experiments, as the models achieve high perturbation fidelity when checking the generated samples with models with different architectures.
> To further alleviate this concern, ConLUX can also use multiple generative models as discussed in Section 4.5, which additionally reduces the risk of perturbation bias.
>
> **[W2] The Impact of Extracted Concepts' Quality on ConLUX Explanations**
> We addressed this by showing the robustness of ConLUX to imperfect concepts.
> As we showed in our response to **Weakness 4 of Reviewer LBfo**, ConLUX is robust to concept correlations and incompleteness.
> We have also added this discussion to the appendix of the updated paper.
>
> **[W3] Running Overhead of ConLUX**
> We provided a detailed response to this concern in our response to **Weakness 1 and Question 1 of Reviewer LBfo**.
> We have also added the discussion about running overhead to the main paper.

---

### Author Response · Authors · 2025-12-03
**Summary for Area Chair**

Dear Area Chair,

We were deeply shocked by the data breach and sincerely sympathize with you for the additional burden it has imposed. To facilitate navigation of our discussions with the reviewers, we have prepared a concise summary of the key points below; however, please feel free to refer to the original threads for full context.

**Reviewer 1sQ1**
They did not reply to us directly but **maintained a positive rating (8) for our work**.

**Reviewer Hsv2**
We continued discussions regarding their concerns about the ConLUX-Anchors explanations. They explicitly stated the condition for improving their rating in their response titled ["Thanks for the rebuttal"](https://openreview.net/forum?id=1bKvM7Ay9G&noteId=smim7IhGxx):

> "I would like to improve my score depending upon the results on weakness 1."

After the cessation of the discussion, we provided the additional experimental results they requested in our most recent response. These results verified that the additional regions introduced by ConLUX-Anchors indeed make a significant contribution to the prediction.

We believe Reviewer Hsv2 will improve their rating accordingly, as they previously promised.

**Reviewer LBfo**
They expressed satisfaction with our responses and **increased their rating to 6** accordingly.

**Reviewer aiQU**
They also expressed satisfaction with our responses and **increased their rating to 6** accordingly.


**Updates on the Paper**
We have also incorporated their valuable suggestions into the revised paper.
Specifically, the reviewers' concerns are mainly about the running overhead of ConLUX and the concept-to-feature mapping process. We have added these discussions to the main paper.
For other valuable discussions, we have also included them in the appendix.
For easy identification, we highlighted the changes in blue in the updated paper.


Best regards,
The authors of the paper "Concept-Based Local Unified Explanations"

---

### Meta-Review · Area_Chair_iWPc · 2025-12-24

**Summary:**

The paper proposed ConLUX, a framework for raising model-agnostic explanation methods from the feature to the concept level. At the same time, the proposed framework enables the generation of different type of explanations, i.e. attribution maps, counterfactuals and sufficient conditions.

The reviewers acknowledged the novelty of the proposed method on operating at the concept level and found the compatibility of the proposed method to different explanation methods and data modalities a strong property to have.

On the down side, a general concern was the added computational costs introduced by the need of a LLM or a diffusion process which, as shown in results reported in the rebuttal, is significant. Moreover, while the rebuttal stated that a large models could be run on a consumer-grade GPU, there is no guarantee that the considered model size and hardware setup will generalize to the explanation of problems addressing other tasks different than the ones considered in the manuscript.

Related to the point above, in response to Reviewer aiQU, the rebuttal indicated that part of the “minimal user effort” that characterizes the proposed method is related to minimal data annotation. However this is achieved by the the need for a of a LLM or a diffusion process that suffer from what is stated above.

An additional major concern was related to the effect introduced by the way the proposed method replaced concepts in the inputs being processed and its comparison w.r.t. replacing it by background (Reviewer-Hsv2). The choice of the procedure to follow is not a subjective choice, and  as accurately stated by the reviewer this will have an effect on the result.

Finally, as is visible in the discussion, in more than one occasion there seem to be misunderstandings on what is presented on the paper. This suggests that the content of the manuscript might not be sufficiently clear as to communicate details that would facilitate reproducibility.

**Reviewer Concerns:**

Addressed Concerns:

- Reviewer 1sQ1:

    - w2: The impact of extracted concepts

    - w3:explicit report of runtime and computational budget

- Reviewer Hsv2:

    - w2: concern on the use of object-centric concepts on the experiments based on image models was addressed.

    - w4: concern on computation costs were addressed.

- Reviewer LBfo:

    - Stated most of his concerns were properly addressed. Increased his/her scores accordingly.

- Reviewer aiQU:

    - Q1: the need for the different types of explanations supported by the  proposed method was properly motivated.

    - Q2: the novelty put forward by the proposed method was clarified as that of moving model-agnostic explanation methods to operate at the concept level.

    - Q3: The effort required on the user side was explicitly stated.


Outstanding Concerns:

- ´Reviewer 1sQ1:

    - Q1: the answer here seems to suggest that the proposed pipeline has the flexibility to accommodate such functionality and an existing effort is pointed to. Unfortunately no more concrete details on the question are provided.

    - Q2: the answers given are problems on themselves, i.e. how to ensure proper prompt diversity is in place, adding in-context learning and/or RAG approaches. w1:

- Reviewer Hsv2:

    - The concern on the actual effect of the replacing concepts with new concepts vs. removing concepts  when producing counterfactuals (raised as part of weakness 1 and 3), was to a good extent left unaddressed. The authors initially hinted at a misunderstanding on the definition of counterfactuals and the procedures to extract them. Then (response Nov. 22), the authors indicated that removal of a concept would require a generative model (which is not accurate).   Finally, after some discussions (response Nov. 27), the authors  stated that whether one of these two procedures is to be followed is a subjective manner.

- Reviewer LBfo:

    - N.A.

- Reviewer aiQU:

    - N.A.

**Reviewer Scores:**

- Reviewer 1sQ1: A follow up on Q1 and request concrete evidence on the capabilities of the proposed method to handle concept hierarchies. Interaction would had allowed the reviewer to inquire further on the “how” of the answers given in the rebuttal. From the weakeness and questions put forward by the reviewer, and considering the discussion with the other reviewers, I doubt the reviewer would had increased his/her initial score.

- Reviewer Hsv2: The reviewer did a diligent work following up on the rebuttal/discussion. After several interactions the pointers given to weaknesses 1&3 were somewhat unsatisfactory. I doubt additional interactions would had helped to bring clarity into the matter and lead to a score raise.

- Reviewer LBfo: The reviewer did participate in the discussion. At the end he/she stated that most of his concerns were properly addressed. Increased his/her scores accordingly.

- Reviewer aiQU: Most questions of the reviewer were clarification questions. To a good extent these were properly addressed.  A proper discussion would had allowed the reviewer to further inquire in the novelty of the proposed method, its way of operation (with the diffusion/LLM-extracted concepts) and the replacement of the need for  annotations by the need for a LLM/diffusion process. From what is visible with the discussion with other reviewers, in these fronts the proposed method is weaker. For this reason, it is likely the reviewer will keep his/her initial score at best.

---

### Decision · Program_Chairs · 2026-01-26

Reject